# Tailored collagen binding of albumin-fused hyperactive coagulation factor IX dictates in vivo distribution and functional properties

Kristin Hovden Aaen [1,2,3,12], Maria Francesca Testa[4,12], Jeannette Nilsen[1,2,3], Rebecca Tarantino[4], Cesare Canepari[5,6], Mascia Benedusi[7], Sopisa Benjakul [1,2,3], Mari Nyquist-Andersen [1,2,3], Marie Leangen Herigstad[1,2,3], Alessio Cantore [5,6], Giuseppe Valacchi[8,9,10], Inger Sandlie[11], Francesco Bernardi[4], Mirko Pinotti[4], Alessio Branchini [4] ✉ & Jan Terje Andersen [1,2,3] ✉

The efficacy of hemophilia B (HB) replacement therapy is evaluated by coagulation factor IX (FIX) activity in plasma, although FIX bound to extravascular type IV collagen (Col4) also contributes to efficient hemostasis. Here, we investigated the impact of engineering FIX for improved (K5R) or reduced (K5A) Col4 binding on the pharmacokinetic properties of FIX Padua, fused to human serum albumin ($HSA_{QMP}$) engineered for favorable neonatal Fc receptor (FcRn) engagement. Hyperactive features and extended plasma half-life in human FcRn expressing mice, attributed to FIX Padua and $HSA_{QMP}$ engineering, respectively, was confirmed. In HB mice, $Padua_{KA}$-$HSA_{QMP}$ exhibited negligible extravascular distribution and the highest plasma levels at early time points followed by the steepest decay. Conversely, $Padua_{KR}$-$HSA_{QMP}$ showed increased extravascular distribution and a 3-fold longer functional half-life (80 hours). These findings support the use of $Padua_{KA}$-$HSA_{QMP}$ and $Padua_{KR}$-$HSA_{QMP}$ as hyperactive short- or long-term therapeutics, respectively, with opportunities for tailored HB replacement therapy.

Hemophilia B (HB) patients are treated with recombinant coagulation factor IX (FIX) as replacement therapy[1]. Several approaches have been explored to extend the short plasma half-life of FIX (18–22 h in humans)[2,3] to enable less frequent dosing, and thus reduce the burden of treatment. This has been achieved by conjugation of FIX to polyethylene glycol (Refixia, Novo Nordisk)[4,5], or by genetic fusion to the Fc region of human immunoglobulin G 1 (IgG1; Alprolix, Bioverativ Therapeutics)[6–9] or to full-length human serum albumin (HSA;

Idelvion, CSL Behring)[10–12]. Fusion of FIX to an IgG1 Fc or to HSA has extended its plasma half-life to 3–4 days in humans[7,8,12–14], due to engagement of the neonatal Fc receptor (FcRn) that rescues IgG and HSA from intracellular degradation via a pH-dependent recycling mechanism[15–20], which provide the endogenous ligands with 3 week-long half-lives in humans[21,22]. We recently reported on an extended half-life (EHL) FIX-HSA fusion protein composed of the hyperactive FIX R338L (Padua) variant and an engineered HSA variant (E505Q/T527M/

[1]Institute of Clinical Medicine, Department of Pharmacology, University of Oslo, Oslo, Norway. [2]Department of Immunology, Oslo University Hospital Rikshospitalet, Oslo, Norway. [3]Precision Immunotherapy Alliance (PRIMA), University of Oslo, Oslo, Norway. [4]Department of Life Sciences and Biotechnology, University of Ferrara, Ferrara, Italy. [5]San Raffaele Telethon Institute for Gene Therapy, IRCCS San Raffaele Scientific Institute, Milan, Italy. [6]Vita-Salute San Raffaele University, Milan, Italy. [7]Department of Neuroscience and Rehabilitation, University of Ferrara, Ferrara, Italy. [8]Department of Environmental Sciences and Prevention, University of Ferrara, Ferrara, Italy. [9]Department of Neurosciences and Rehabilitation; Animal Science Department, NC Research Campus, Plants for Human Health Institute, NC State University, Kannapolis, NC, USA. [10]Department of Food and Nutrition, Kyung Hee University, Seoul, Republic of Korea. [11]Department of Biosciences, University of Oslo, Oslo, Norway. [12]These authors contributed equally: Kristin Hovden Aaen, Maria Francesca Testa. ✉ e-mail: brnlss@unife.it; j.t.andersen@medisin.uio.no

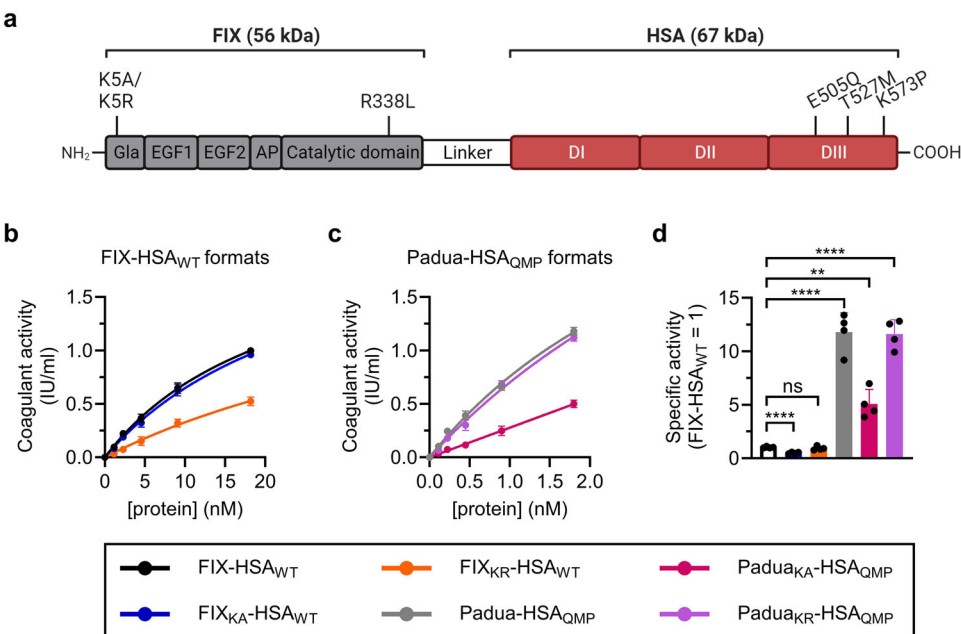

**Fig. 1 | Design and activity profiles of FIX-HSA variants with modifications in both fusion partners. a** Four fusion proteins were designed with FIX genetically fused to HSA via a cleavable linker. FIX was modified in position 5 of the Gla domain (K5A or K5R) and in position 338 in the catalytic domain (R338L; Padua). HSA was either unmodified (WT) or engineered with three amino acid substitutions (E505Q/T527M/K573P; QMP) in DIII. Gla, γ-carboxyglutamic acid domain; EGF1/EGF2, epidermal growth factor like domains 1 and 2; AP, activation peptide. Coagulant activity of FIX measured by aPTT-based assays in FIX-deficient plasma supplemented with pure monomeric fractions of (**b**), FIX-HSA$_{WT}$ or (**c**), Padua-HSA$_{QMP}$. Data represent the mean ± SD of technical duplicates. **d** Specific clotting activity of each FIX-HSA variant. The data represent the mean ± SD of technical quadruplets, where specific activity is calculated as the ratio between activity and protein levels (FIX-HSA$_{WT}$ = 1). Statistical significance was tested by unpaired two-tailed Student's $t$ tests with 95% confidence level ($n$ = 4; ns $p$ = 0.8792; **$p$ = 0.0010, ****$p$ < 0.0001). FIX-HSA$_{WT}$, black; Padua-HSA$_{QMP}$, gray; FIX$_{KA}$-HSA$_{WT}$, blue; FIX$_{KR}$-HSA$_{WT}$, orange; Padua$_{KA}$-HSA$_{QMP}$, pink; Padua$_{KR}$-HSA$_{QMP}$, purple. Source data are provided as a Source Data file. **a** Created in BioRender. Hovden Aaen, K. (2025), https://BioRender.com/auh41ln.

K573P; HSA$_{QMP}$) with improved pH-dependent FcRn binding, exhibiting an additional 2.5-fold extended plasma half-life in human FcRn (hFcRn) transgenic mice[23–25].

In addition to circulating in blood, FIX binds extravascular type IV collagen (Col4)[26–32]. This is supported by the observations in vivo that i) in gene therapy experiments, FIX secreted from transduced muscle cells remains bound to the surrounding matrix[33], ii) 50–80% of injected FIX rapidly disappear from plasma[34,35], and iii) FIX effectively prevents bleeding even at plasma levels below 1%, while maintaining adequate trough levels in HB prophylaxis[36–44]. Thus, infused FIX protects from bleeding for longer than predicted by its plasma half-life, with extravascular Col4-bound FIX playing an important hemostatic function. Moreover, HB patients who receive regular FIX infusions require less over time to achieve the targeted plasma level, suggesting that the extravascular depot becomes saturated[37,39]. Importantly, the HB cross-reactive material (CRM) status, associated with either undetectable FIX (CRM⁻) or circulating dysfunctional (d)FIX (CRM⁺), may affect to what degree an injected FIX enters the extravascular depot and, thus, the resulting pharmacokinetic (PK) profile[36,38,42,43,45,46].

The binding of FIX to Col4 relies on amino acid residues 3–11 in the N-terminal γ-carboxyglutamic acid (Gla) domain[30,47,48]. Interestingly, replacement of a lysine residue at position 5 with an alanine (K5A) reduces the binding to Col4, while an arginine (K5R) strengthens it[30,31,35,49]. Furthermore, it has been shown that infusion of K5R FIX gives HB mice prolonged and enhanced protection against bleeding, compared to the wild-type (WT) and K5A FIX variants, even when undetectable in the circulation[41,50].

Although it is already known that infused FIX, both free and fused to HSA, distributes to the extravascular space[34,51–53], the impact of Col4 binding on the PK profile of FIX has been poorly addressed, especially in the context of EHL fusion proteins. Thus, we investigated the functional effects of K5 engineering in FIX-HSA fusion proteins in three mouse models mimicking different CRM-statuses and provide evidence for the contributions of the K5A and K5R substitutions on FIX biodistribution and functional half-life, with implications for HB treatment.

## Results

### FIX-HSA fusions retain clotting activity after K5 engineering

To produce FIX-HSA fusion proteins with modulated Col4 binding properties, we designed expression vectors encoding either WT FIX or hyperactive Padua (R338L)[23] variants fused in-frame with full-length HSA (FIX-HSA$_{WT}$, FIX$_{KA}$-HSA$_{WT}$, FIX$_{KR}$-HSA$_{WT}$) or an engineered HSA variant (HSA$_{QMP}$)[24] (Padua-HSA$_{QMP}$, Padua$_{KA}$-HSA$_{QMP}$, Padua$_{KR}$-HSA$_{QMP}$) (Fig. 1a)[25]. Production in a human cell-based system yielded monomeric and pure fractions of all variants (Supplementary Figs. 1, 2).

To investigate the coagulant activity of the designed FIX-HSA fusions, aPTT-based and chromogenic activity assays were performed. The results showed dose-dependent coagulant activities, with normal (Fig. 1b) or hyperactive (Fig. 1c) profiles, dependent on the absence or presence of the Padua amino acid substitution (Supplementary Fig. 3). The introduction of the K5 substitutions resulted in either unaltered (K5R; 0.9-fold of FIX-HSA$_{WT}$) or reduced (K5A; 0.5-fold of FIX-HSA$_{WT}$) activity, while their combination with Padua gave rise to hyperactive features (averaging 9.5-fold higher than FIX-HSA$_{WT}$; Fig. 1b–d). Moreover, upon incubation of FIX with plasma-derived activated FXI (pdFXIa), HSA was released from Padua$_{KA}$-HSA$_{QMP}$ and Padua$_{KR}$-HSA$_{QMP}$ via its cleavable linker at comparable rates (Supplementary Fig. 4). To examine whether the reduced activity observed after introducing K5A could be related to a modified phospholipid binding, an ELISA-based assay was performed where vesicular phospholipids were coated in wells, before a

concentration gradient of Padua-HSA$_{QMP}$, Padua$_{KA}$-HSA$_{QMP}$, or Padua$_{KR}$-HSA$_{QMP}$ was added, and the HSA region of the fusion proteins was detected. The results indicated that the fusions bound phospholipids differently, with the strongest binding observed for the K5R-engineered fusion (Padua$_{KR}$-HSA$_{QMP}$), followed by that unmodified in position 5 (Padua-HSA$_{QMP}$), and lastly the K5A-engineered fusion (Padua$_{KA}$-HSA$_{QMP}$; Supplementary Fig. 5).

### K5 engineering in FIX-HSA does not affect FcRn binding

To study the ability of the FIX-HSA fusions to engage soluble FcRn (Supplementary Fig. 6) in a pH-dependent manner, an ELISA-based binding assay was performed (Fig. 2a). All HSA$_{QMP}$-containing variants bound strongly to hFcRn compared with the HSA$_{WT}$ counterparts at pH 5.5, while no binding was detected at pH 7.4 (Fig. 2b; Supplementary Fig. 7). This was also the case for binding to mFcRn, although the binding responses were lower (Fig. 2c), in line with previous observations[54–56]. Importantly, no effect of K5 engineering on binding to either FcRn species was observed (Fig. 2b, c).

To determine the binding kinetics, surface plasmon resonance (SPR) was performed (Fig. 2d). At pH 5.5, similar equilibrium constants ($K_D$) of $176.7 \pm 11.3$ nM and $190.3 \pm 17.6$ nM were derived for FIX$_{KA}$-HSA$_{WT}$ and FIX$_{KR}$-HSA$_{WT}$, respectively, towards hFcRn (Fig. 2e, f; Table 1). In contrast, hFcRn dissociated more slowly from the HSA$_{QMP}$-containing variants with $K_D$ values of $0.2–0.3$ nM (Fig. 2g, h; Table 1). Again, no influence of K5 engineering was measured (Fig. 2e–h). No or negligible binding was observed at pH 7.4 (Supplementary Figs. 8a–d, 9a–d).

Moreover, mFcRn bound the HSA$_{WT}$-containing variants poorly, whereas the QMP substitutions increased the binding at pH 5.5, resulting in $K_D$ values of $44.6–54.9$ nM (Fig. 2i–l; Table 1). No detectable binding was observed at pH 7.4 (Supplementary Figs. 8e–h, 9e–h). Thus, the HSA$_{QMP}$-containing variants showed strict pH-dependent binding against both FcRn species, with about 200-fold stronger binding toward the human receptor.

### FIX-HSA variants are rescued from intracellular degradation by FcRn

To investigate FcRn-dependent rescue of the FIX-HSA variants from intracellular degradation, a human endothelial cell-based recycling assay (HERA)[57] was performed with HMEC-1-hFcRn cells (Fig. 2m). After the uptake phase, 3- to 5-fold more of the HSA$_{QMP}$-containing variants were detected inside the cells than their HSA$_{WT}$ counterparts (Fig. 2n; Supplementary Fig. 10a), while more than 10-fold more was detected in the recycling medium (Fig. 2o; Supplementary Fig. 10b), supporting that the QMP substitutions enhanced FcRn-mediated rescue from intracellular degradation. Importantly, neither fusion of FIX to HSA nor the K5 substitutions affected cellular handling (Fig. 2n, o; Supplementary Figs. 10, 11). Thus, K5 engineering in FIX does not affect FcRn-mediated cellular recycling of the HSA-fused variants.

### No effect of K5 engineering on plasma half-life in hFcRn expressing mice

To investigate whether K5 engineering affects the PK properties of FIX-HSA in the most clinically relevant mouse model for studies of HSA, we used Tg32 mice that are transgenic for hFcRn and do not express the mouse counterpart[58]. These mice express normal mFIX levels ($250.0 \pm 20.0$ ng/mL; Supplementary Fig. 12), and thus mimic type II (CRM$^+$) patients in terms of circulating FIX protein (Fig. 3a).

Twenty-four hours post-administration (Fig. 3b), more than 3.5-fold higher plasma concentrations were detected of the HSA$_{QMP}$-containing variants than of those with HSA$_{WT}$, at an average of 14.4 and 4.0 µg/mL, respectively (Fig. 3c; Supplementary Fig. 13). Furthermore, FIX containing either K5A or K5R, when fused to HSA$_{WT}$, showed plasma half-lives of $23.9 \pm 5.6$ and $20.0 \pm 1.3$ h, respectively, comparable with that of FIX-HSA$_{WT}$ ($27.4 \pm 1.8$ h) (Fig. 3d; Supplementary

Fig. 13). In stark contrast, the HSA$_{QMP}$-containing fusions showed more than 3-fold extended plasma half-lives (Padua$_{KA}$-HSA$_{QMP}$, $75.9 \pm 19.2$ h; Padua$_{KR}$-HSA$_{QMP}$, $77.6 \pm 20.0$ h), regardless of A or R in position 5 of FIX (Fig. 3d).

When applying the measured protein concentrations to gPKPDsim[59], comparable mean residence time (MRT) values were measured for the HSA$_{WT}$-containing variants (1.6–1.9 days), while those for the HSA$_{QMP}$-containing variants were approximately 5- and 6-fold longer (Padua$_{KA}$-HSA$_{QMP}$, 7.9 days; Padua$_{KR}$-HSA$_{QMP}$, 9.6 days) (Table 2). The clearance (CL) rate of the HSA$_{WT}$- and HSA$_{QMP}$-containing variants were about 156.6 mL/d/kg and 19.6 mL/d/kg on average (Table 2).

Thus, QMP engineering is the main driver of the half-life extension of FIX-HSA in the context of the human receptor and normal levels of endogenous FIX, whereas K5 engineering appears to exert negligible effects.

### QMP extends the plasma half-life of FIX-HSA in WT mice

To address how the differences in binding to the FcRn species affect the PK properties of the FIX-HSA fusions, we performed a PK study in Balb/c mice (Fig. 3e). These mice express mFcRn and normal mFIX levels (FIX$^{plus}$ mice, $320 \pm 30.0$ ng/mL; Supplementary Fig. 12), and thus, also mimic type II CRM$^+$ patients. As the HSA$_{WT}$-containing variants bind mFcRn poorly, we took advantage of the HSA$_{QMP}$-containing fusions, and included FIX-HSA$_{WT}$ for comparison.

Twenty-four hours post-administration, more than 2.5-fold higher plasma concentrations of the HSA$_{QMP}$-containing variants (4.9 µg/mL) compared to FIX-HSA$_{WT}$ (1.8 µg/mL) were detected, on average (Fig. 3f; Supplementary Fig. 14). Furthermore, FIX-HSA$_{WT}$ showed a plasma half-life of $14.1 \pm 1.5$ h, 2-fold shorter than those measured for the HSA$_{QMP}$-containing variants (Fig. 3g). Although not significantly, a slightly shorter plasma half-life was measured for Padua$_{KA}$-HSA$_{QMP}$ ($30.6 \pm 4.9$ h) compared to Padua-HSA$_{QMP}$ ($39.2 \pm 6.8$ h) and Padua$_{KR}$-HSA$_{QMP}$ ($35.7 \pm 3.8$ h) (Fig. 3g).

QMP engineering of HSA resulted in about 2-fold longer MRT, with the longest observed for Padua$_{KR}$-HSA$_{QMP}$ (1.9 days), followed by Padua$_{KA}$-HSA$_{QMP}$ (1.5 days), Padua-HSA$_{QMP}$ (1.4 days), and lastly, FIX-HSA$_{WT}$ (0.7 days) (Table 2). Accordingly, CL rates at an average of 285.3 and 106.2 mL/d/kg for the HSA$_{WT}$- and HSA$_{QMP}$-containing fusions were measured, respectively (Table 2).

Thus, the QMP substitutions extend the plasma half-life of FIX-HSA in the mFcRn expressing FIX$^{plus}$ mice, whereas the K5A substitution appeared to shorten the plasma half-life of the fusion in this mouse model.

### K5 engineering modulates the PK profile of FIX-HSA in HB mice

To investigate the effect of K5 engineering in the absence of an extravascular FIX depot, we performed a study in HB mice (Fig. 4a)[60]. The HB mice do not express FIX (Supplementary Fig. 12), and hence, mimic CRM$^-$ patients where Col4 is freely accessible to bind injected FIX.

One hour post-administration, Padua$_{KA}$-HSA$_{QMP}$ was detected at an almost 3-fold higher plasma concentration ($24.2 \pm 9.3$ µg/mL) than FIX$_{WT}$-HSA$_{WT}$, Padua-HSA$_{QMP}$, and Padua$_{KR}$-HSA$_{QMP}$, which were found at 8.2 µg/mL, on average (Fig. 4b). After 24 h, the plasma levels were reduced, with Padua$_{KA}$-HSA$_{QMP}$ still being the most prominent variant at $2.2 \pm 0.5$ µg/mL followed by 1.5-fold less of Padua$_{KR}$-HSA$_{QMP}$ ($1.5 \pm 0.5$ µg/mL), comparable to the amount of FIX-HSA$_{WT}$ ($1.6 \pm 0.1$ µg/mL) (Fig. 4c). Padua-HSA$_{QMP}$ was found at slightly lower levels ($0.9 \pm 0.1$ µg/mL) (Fig. 4c). After 96 h, 2-fold more of Padua$_{KR}$-HSA$_{QMP}$ ($0.30 \pm 0.11$ µg/mL) than Padua$_{KA}$-HSA$_{QMP}$ ($0.15 \pm 0.02$ µg/mL) was present in plasma (Fig. 4d), corresponding to 19% and 7% of the-levels measured after 1 day, respectively (Fig. 4e, f). Consequently, Padua$_{KA}$-HSA$_{QMP}$ exhibited the steepest elimination curve (Fig. 4f).

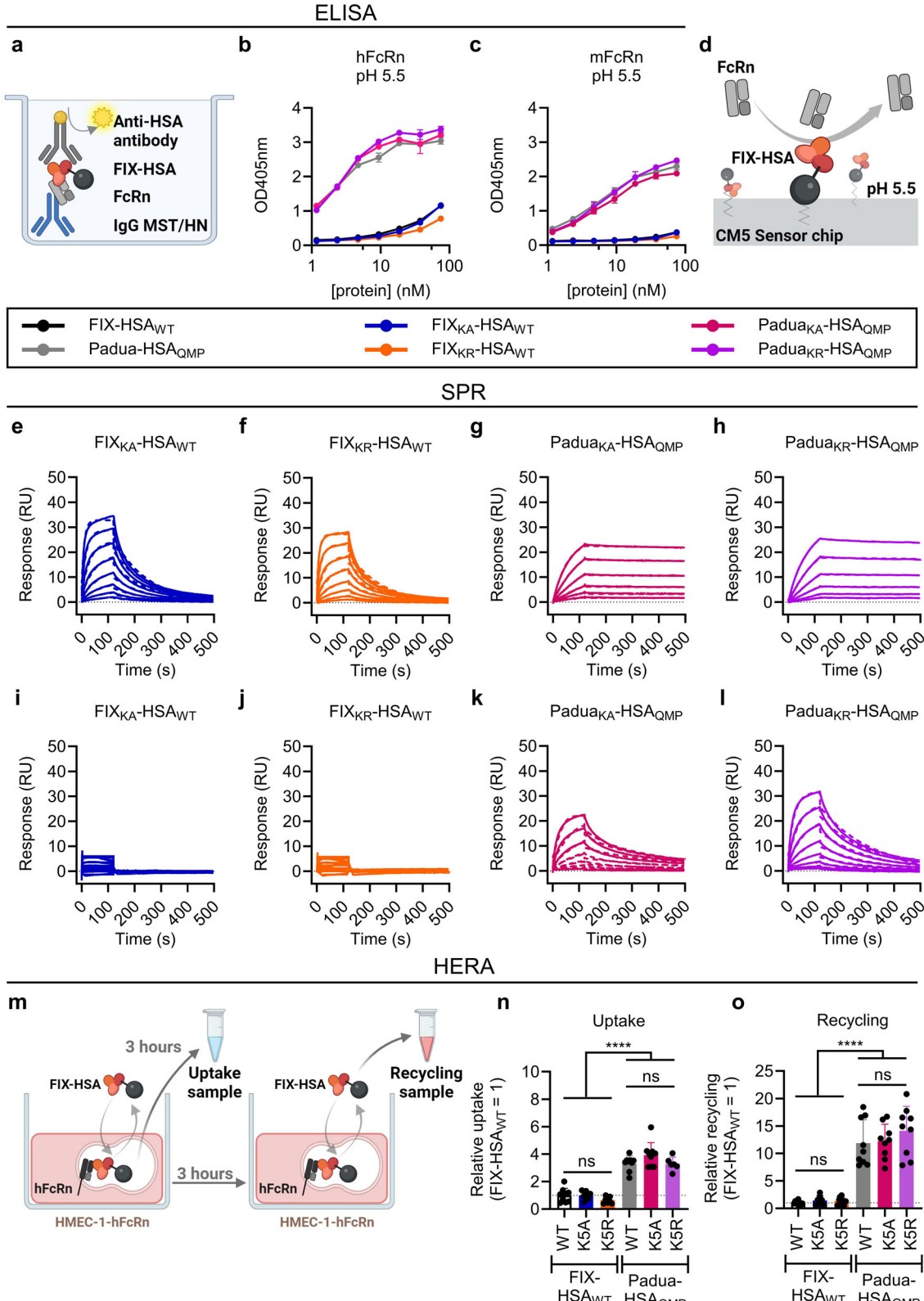

Noteworthy, the plasma concentrations of all four variants were lower than in the FIX^plus mice.

Furthermore, Padua$_{KA}$-HSA$_{QMP}$ showed the shortest plasma half-life (17.7 ± 1.2 h), followed by FIX$_{WT}$-HSA$_{WT}$ (21.9 ± 0.7 h), Padua-HSA$_{QMP}$ (25.0 ± 2.1 h), and lastly, Padua$_{KR}$-HSA$_{QMP}$ (26.0 ± 2.6 h) (Fig. 4g). The gPKPDsim analysis showed that K5R increased the MRT

of the fusion (Padua$_{KR}$-HSA$_{QMP}$, 1.4 days; Padua-HSA$_{QMP}$, 1.2 days), whereas the K5A substitution shortened it by 3-fold (Padua$_{KA}$-HSA$_{QMP}$, 0.4 days) (Fig. 4h; Table 2).

Thus, in mice without endogenous FIX, K5A provides higher plasma concentrations of Padua-HSA$_{QMP}$ shortly after administration, whereas K5R improves its PK profile.

**Fig. 2 | QMP engineering in HSA yields FIX-HSA fusion proteins with favorable FcRn binding properties and FcRn-mediated cellular handling. a** Illustration of the ELISA-based binding assay used to study binding between hFcRn or mFcRn and FIX-HSA variants at pH 5.5. Soluble truncated FcRn (light gray) was captured on IgG1 MST/HN (blue) coated in the well, before FIX-HSA (red/black) was added to the wells, and the HSA region of the fusion protein was detected by an ALP-conjugated anti-HSA antibody (dark gray). **b, c** Results from the ELISA-based hFcRn and mFcRn binding assays performed at pH 5.5. Data represent the mean ± SD of technical duplicates from one representative experiment. **d** Illustration of the SPR experiment used to study the interaction between FcRn (gray) and FIX-HSA (orange/black) performed by immobilizing FIX-HSA fusion proteins on a CM5 sensor chip before injecting a truncated form of soluble FcRn in a concentration gradient. Representative sensorgrams from SPR performed by injecting soluble truncated hFcRn (**e–h**) or mFcRn (**i–l**) over immobilized FIX-HSA variants at pH 5.5, showing one out of three independent experiments performed. The dotted lines show curves fitted by the 1:1 Langmuir binding model. **m** An illustration of the HERA setup designed to study hFcRn-mediated cellular (**n**), uptake and (**o**), recycling of FIX-HSA fusion proteins. Data are presented as relative to FIX-HSA$_{WT}$ (=1, dotted line) and represent the group mean ± SD of three independent experiments performed in technical triplicates (uptake: $n = 9$, except $n = 8$ for FIX$_{KA}$-HSA$_{WT}$ and FIX$_{KR}$-HSA$_{WT}$, $n = 6$ for Padua$_{KR}$-HSA$_{QMP}$; recycling: $n = 9$, except $n = 7$ for FIX$_{KA}$-HSA$_{WT}$). Statistical significance was tested by unpaired two-tailed Student's $t$ tests with 95% confidence level (precise $p$-values are given in Supplementary Table 1). FIX-HSA$_{WT}$, black; Padua-HSA$_{QMP}$, gray; FIX$_{KA}$-HSA$_{WT}$, blue; FIX$_{KR}$-HSA$_{WT}$, orange; Padua$_{KA}$-HSA$_{QMP}$, pink; Padua$_{KR}$-HSA$_{QMP}$, purple. Source data are provided as a Source Data file. **a, d, m** Created in BioRender. Hovden Aaen, K. (2025) https://BioRender.com/auh41ln.

## K5R increases the extravascular presence of FIX-HSA fusions in HB mice

To evaluate the biodistribution of the FIX-HSA variants, we collected the livers, kidneys, lungs, and knee joints at termination of the experiment in HB mice (day 4) and analyzed the corresponding homogenates by ELISA. The highest protein concentrations were detected in the liver, followed by the kidneys, lungs and, lastly, the knee joints (Fig. 4i–l). In the liver, comparable amounts of the fusions unmodified in position 5 were detected (FIX-HSA$_{WT}$ and Padua-HSA$_{QMP}$, 97.6–107.0 ng/mL), while 3-fold less of Padua$_{KA}$-HSA$_{QMP}$ (35.0 ± 5.5 ng/mL) and 2-fold more of Padua$_{KR}$-HSA$_{QMP}$ (231.6 ± 61.5 ng/mL) were quantified (Fig. 4i). Similar differences between the variants were observed in the kidneys (Fig. 4j), lungs (Fig. 4k), and knee joints (Fig. 4l).

To further study the tissue distribution of the fusion proteins, harvested organs were sectioned and stained with primary goat and rabbit polyclonal antibodies specific for human FIX or mouse Col4, respectively, before detection using species-matched secondary antibodies. Immunofluorescent (IF) staining revealed significantly stronger fluorescence intensity in tissues from HB mice injected with Padua$_{KR}$-HSA$_{QMP}$ than with Padua$_{KA}$-HSA$_{QMP}$ (Fig. 4m), while the signals for mouse Col4 were not significantly different (Supplementary Fig. 15). In line with this, immunohistochemistry (IHC) staining confirmed detection of more Padua$_{KR}$-HSA$_{QMP}$ than Padua$_{KA}$-HSA$_{QMP}$ (Supplementary Fig. 16). Again, the tissues showed similar IHC staining of mouse Col4 irrespective of treatment either with the K5A or K5R engineered fusion (Supplementary Fig. 16). Notably, treating the tissues with only the secondary antibodies did not give any signal (Supplementary Fig. 17).

Thus, whereas K5A greatly reduces the extravascular presence of Padua-HSA$_{QMP}$, K5R significantly increases its distribution to the liver, kidneys, lungs, and knee joints, at sites with mouse Col4 expression.

## K5R grants sustained enhancement in plasma FIX activity in HB mice

To investigate the functional activity level of the FIX-HSA variants over time, the plasma samples from the HB mice were evaluated in aPTT-based activity assays. One hour post-administration, the activity level of Padua$_{KA}$-HSA$_{QMP}$ (1232.6 ± 446.1% of FIX-HSA$_{WT}$) was about 2-fold higher than that of Padua$_{KR}$-HSA$_{QMP}$ (585.3 ± 245.7%) and Padua-HSA$_{QMP}$ (715.9 ± 84.1%) (Fig. 5a). After 24 h, all three Padua-containing fusions showed comparable activity levels, approximately 6- to 8-fold higher than that of FIX-HSA$_{WT}$ (Fig. 5b). At the final 96-hour time point, Padua$_{KR}$-HSA$_{QMP}$ (339.5 ± 58.9%) demonstrated the highest activity level, corresponding to 16-fold higher activity than FIX-HSA$_{WT}$ (20.7 ± 4.1 %) (Fig. 5c). In contrast, Padua$_{KA}$-HSA$_{QMP}$ (48.2 ± 9.2%) showed the lowest activity level of all the Padua-containing variants at this time point (Fig. 5c), coupled with the most rapid decline in activity over time (Fig. 5d). Overall, all the Padua-containing fusions exhibited superior activity levels in plasma at all time points, compared to FIX-HSA$_{WT}$ (Fig. 5e).

When comparing the residual activity at each time point compared to that at day 1, Padua-HSA$_{QMP}$ and FIX-HSA$_{WT}$ followed a comparable decline over time, consistent with their clearance curves (Fig. 5f), while maintaining the same relative differences in activity levels (Fig. 5d). At day 4, these variants showed 20.9% and 17.1% of their initial activity, respectively (Fig. 5f). Moreover, Padua$_{KR}$-HSA$_{QMP}$ showed the highest residual activity (43.9%) among all variants at day 4, while Padua$_{KA}$-HSA$_{QMP}$ exhibited the most dramatic decline, reaching 7.6% of its initial activity (Fig. 5f).

Next, when applying the functional activity data to gPKPDsim, an almost 3-fold extended functional half-life was revealed for Padua$_{KR}$-HSA$_{QMP}$ (80.3 ± 14.3 h) compared to that of Padua-HSA$_{QMP}$ (30.6 ± 8.0 h), whereas a 1.5-fold shorter functional half-life was estimated for Padua$_{KA}$-HSA$_{QMP}$ (22.1 ± 6.1 h) (Fig. 5g). Moreover, longer MRT values were obtained for the FIX-HSA fusions when calculated based on their activity levels rather than their plasma concentrations (FIX-HSA$_{WT}$, 1.8 versus 0.7 days; Padua-HSA$_{QMP}$, 1.9 versus 1.2 days; Padua$_{KR}$-HSA$_{QMP}$, 4.4 versus 1.4 days; Padua$_{KA}$-HSA$_{QMP}$, 0.9 versus 0.4 days) (Fig. 5h). Regardless, the K5A and K5R substitutions provided Padua-HSA$_{QMP}$ with more than 2-fold shorter or longer MRT, respectively.

To evaluate the ability of the fusion protein to promote hemostasis in vivo, a tail clip assay was performed on HB mice 4 days after an injection of Padua$_{KR}$-HSA$_{QMP}$. The results showed an inverse correlation between blood loss and plasma activity levels (Fig. 5i), confirming

## Table 1 | SPR-derived kinetic constants for the interaction between FIX-HSA variants and hFcRn or mFcRn at pH 5.5

| FIX-HSA variant[a] | $K_a$ (10$^4$ s$^{-1}$) | $K_d$ (10$^{-4}$ s$^{-1}$) | $K_D$ (nM)[b] |
|---|---|---|---|
| hFcRn | | | |
| FIX$_{KA}$-HSA$_{WT}$ | 4.4 ± 0.4 | 76.9 ± 2.5 | 176.7 ± 11.3 |
| FIX$_{KR}$-HSA$_{WT}$ | 4.4 ± 0.5 | 82.7 ± 0.9 | 190.3 ± 17.6 |
| Padua$_{KA}$-HSA$_{QMP}$ | 33.4 ± 3.7 | 0.6 ± 0.0 | 0.2 ± 0.1 |
| Padua$_{KR}$-HSA$_{QMP}$ | 27.1 ± 1.8 | 0.7 ± 0.1 | 0.3 ± 0.0 |
| mFcRn | | | |
| FIX-HSA$_{WT}$ | NA[c] | NA | NA |
| FIX$_{KA}$-HSA$_{WT}$ | NA | NA | NA |
| FIX$_{KR}$-HSA$_{WT}$ | NA | NA | NA |
| Padua-HSA$_{QMP}$ | 11.6 ± 4.1 | 51.2 ± 17.1 | 44.6 ± 1.9 |
| Padua$_{KA}$-HSA$_{QMP}$ | 5.45 ± 0.3 | 28.0 ± 4.3 | 51.7 ± 10.6 |
| Padua$_{KR}$-HSA$_{QMP}$ | 7.3 ± 2.0 | 42.9 ± 27.8 | 54.9 ± 20.5 |

$K_a$ association rate constant, $K_d$ dissociation rate constant, $K_D$ equilibrium dissociation constant.
[a] FIX-HSA variants were immobilized on CM5 chips and serial dilutions of either the hFcRn or mFcRn were injected at pH 5.5.
[b] Kinetic rate constants, representing the mean ± SD of three independent runs, were obtained using a simple first-order 1:1 Langmuir binding model.
[c] NA not acquired due to fast kinetics.

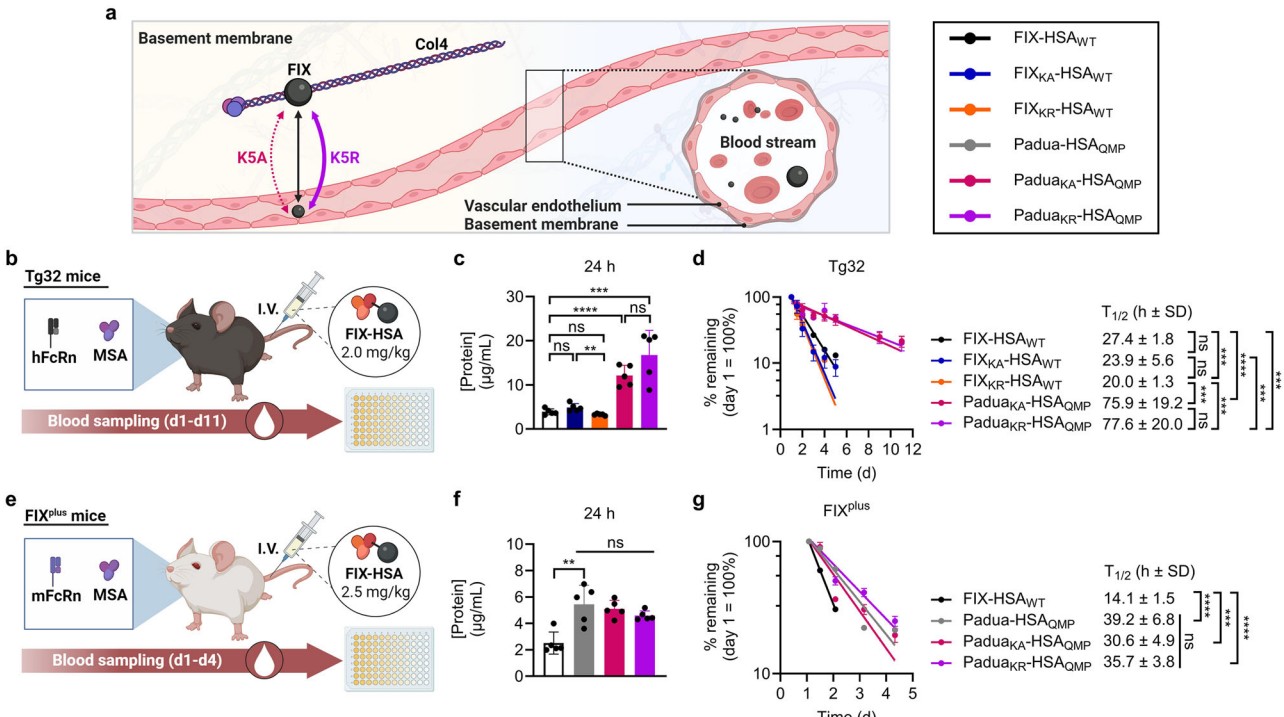

**Fig. 3 | QMP engineering extends the plasma half-life of FIX-HSA. a** Illustration of the extra- and intra-vascular storages of FIX. Infused and endogenous FIX may relocate from the circulation to the extravascular space where it associates with Col4. The K5A and K5R substitutions in FIX weakens or strengthens its interaction with Col4, respectively. **b** Illustration of the half-life study in Tg32 mice where 2 mg/kg of FIX-HSA were administered by I.V. injection. Protein quantification was performed by ELISA. The results are presented as (**c**), protein concentration 24 h after administration and (**d**), percentage remaining in plasma at each time point compared to day 1 (= 100%), with plasma half-life values to the right, and represent the group mean ± SD of biological replicates ($n = 5$). **e** Illustration of the half-life study in FIX$^{plus}$ Balb/c mice where 2.5 mg/kg of FIX-HSA was administered by I.V.

injection. Protein quantification was performed by ELISA. The results are presented as (**f**), protein concentration 24 h after administration and (**g**), percentage remaining in plasma at each time point compared to day 1 (= 100%), with plasma half-life values to the right, and represent the group mean ± SD of biological replicates ($n = 5$). Statistical significance was tested by unpaired two-tailed Student's $t$ tests with 95% confidence level ($n = 5$; precise $p$-values are given in Supplementary Tables 2–5). FIX-HSA$_{WT}$, black; FIX$_{KA}$-HSA$_{WT}$, blue; FIX$_{KR}$-HSA$_{WT}$, orange; Padua-HSA$_{QMP}$, gray; Padua$_{KA}$-HSA$_{QMP}$, pink; Padua$_{KR}$-HSA$_{QMP}$, purple. Source data are provided as a Source Data file. **a**, **b**, **e** Created in BioRender. Hovden Aaen, K. (2025) https://BioRender.com/auh41ln.

**Table 2 | PK and PD parameters of FIX-HSA fusion proteins in different mouse models analyzed using gPKPDsim in MatLab**

| Protein | Dose (mg/kg) | Route | AUC (µg*d/mL) | $C_{max}$ (µg/mL) | CL (mL/d/kg) | MRT (d) | $V_{ss}$ (mL/kg) | $T_{1/2}$ (d) | $R^2$ |
|---|---|---|---|---|---|---|---|---|---|
| Tg32$^{α}$ | | | | | | | | | |
| FIX-HSA$_{WT}$ | 2 | I.V. | 11.9 | 3.9 | 161.2 | 1.7 | 274.2 | 1.3 | 0.971 |
| FIX$_{KA}$-HSA$_{WT}$ | 2 | I.V. | 13.8 | 4.9 | 139.9 | 1.9 | 269.4 | 2.5 | 0.963 |
| FIX$_{KR}$-HSA$_{WT}$ | 2 | I.V. | 11.0 | 3.3 | 168.7 | 1.6 | 263.3 | 2.5 | 0.985 |
| Padua$_{KA}$-HSA$_{QMP}$ | 2 | I.V. | 44.5 | 12.8 | 23.2 | 7.9 | 178.7 | 5.8 | 0.865 |
| Padua$_{KR}$-HSA$_{QMP}$ | 2 | I.V. | 68.0 | 16.7 | 16.0 | 9.6 | 134.0 | 7.4 | 0.485 |
| FIX$^{plus\ β}$ | | | | | | | | | |
| FIX-HSA$_{WT}$ | 2.5 | I.V. | 8.8 | 9.1 | 285.3 | 0.7 | 198.1 | 0.7 | 0.992 |
| Padua-HSA$_{QMP}$ | 2.5 | I.V. | 21.9 | 16.7 | 109.1 | 1.4 | 154.3 | 1.1 | 0.925 |
| Padua$_{KA}$-HSA$_{QMP}$ | 2.5 | I.V. | 22.2 | 18.8 | 103.3 | 1.5 | 152.9 | 1.5 | 0.919 |
| Padua$_{KR}$-HSA$_{QMP}$ | 2.5 | I.V. | 20.5 | 15.5 | 106.1 | 1.9 | 204.9 | 1.9 | 0.958 |
| HB mice$^{γ}$ | | | | | | | | | |
| FIX-HSA$_{WT}$ | 2.5 | I.V. | 8.8 | 11.8 | 279.0 | 0.7 | 193.6 | 1.0 | 0.980 |
| Padua-HSA$_{QMP}$ | 2.5 | I.V. | 4.4 | 5.7 | 514.8 | 1.2 | 570.6 | 1.7 | 0.952 |
| Padua$_{KA}$-HSA$_{QMP}$ | 2.5 | I.V. | 15.9 | 24.2 | 178.8 | 0.4 | 84.03 | 0.8 | 0.987 |
| Padua$_{KR}$-HSA$_{QMP}$ | 2.5 | I.V. | 6.5 | 7.1 | 387.8 | 1.4 | 517.5 | 1.7 | 0.993 |

$^{α}$Tg32, homozygous hFcRn transgenic Tg32 mice.
$^{β}$FIX$^{plus}$, Balb/c mice.
$^{γ}$HB mice, *F9* KO mice.
*AUC* Area under the curve from time 0 to the last measured time point, *$C_{max}$* Maximum plasma concentration, *CL* Clearance rate, *MRT* Mean residence time, *$V_{ss}$* Volume of distribution at steady state, *$T_{1/2}$* Terminal half-life, *$R^2$* Coefficient of determination.

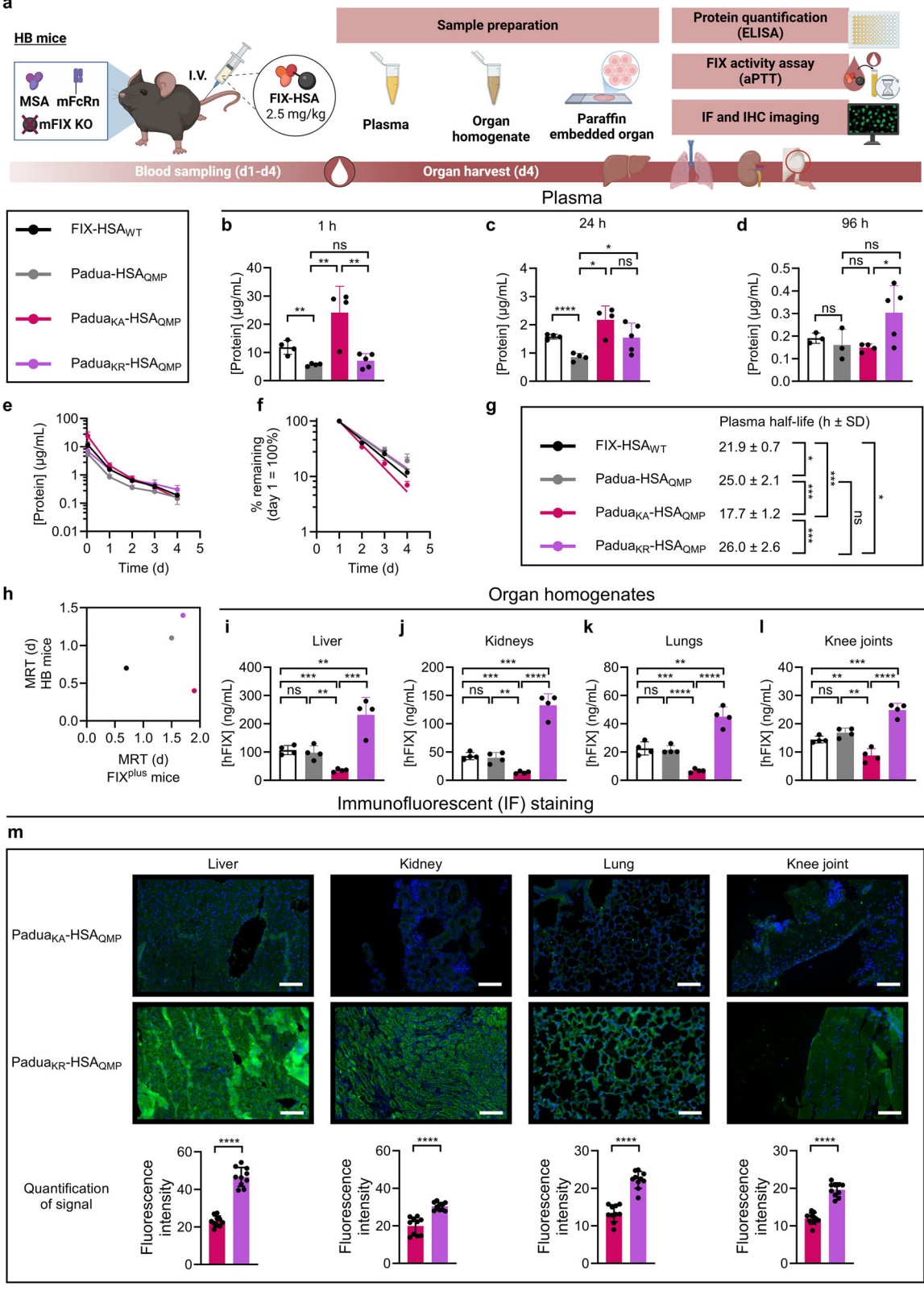

that the Padua_{KR}-HSA_{QMP} present in blood efficiently promoted hemostasis.

Thus, K5R in Padua-HSA_{QMP} extends both the plasma and functional half-life of the fusion protein, while K5A accelerates its elimination following an early peak in plasma level (Table 3). The contribution of each engineering step on the PK profile of the fusion proteins in different mouse models is summarized in Supplementary Table 12.

## Discussion

The first generation of EHL FIX products has represented a major improvement in HB treatment[61–64]. However, as their plasma half-lives

**Fig. 4 | K5R engineering of FIX-HSA increases plasma half-life and extravascular presence in HB mice. a** Illustration of the study in HB mice where 2.5 mg/kg of FIX-HSA was administered by I.V. injection. At termination (day 4), lungs, kidneys, liver, and knee joints were harvested and processed to homogenates, or paraffin embedded and prepared on slides before IF or IHC staining. Blood processed to plasma and tissue homogenates were analyzed by ELISA for protein quantification, and coagulant activity was measured in plasma by an aPTT-based assay. **b–d** Protein concentration in plasma 1 h, 24 h, and 96 h post-administration. Data are presented as the group mean ± SD of biological replicates ($n = 4–5$). Elimination curves showing (**e**), protein concentration remaining in plasma over time and (**f**), percentage of protein remaining in plasma at each time point compared to day 1 (= 100%), presented as the mean ± SD of biological replicates ($n = 4–5$). **g** Plasma half-life values shown as the group mean ± SD of biological replicates ($n = 4–5$). **h** MRT values in FIX[plus] Balb/c mice (x-axis) and HB mice (y-axis) analyzed in gPKPDsim.

**i–l** Homogenates of livers, kidneys, lungs, and knee joints sampled at termination were analyzed by ELISA for protein quantification. Data represent the group mean ± SD of biological replicates ($n = 4–5$). **m** IF stained sections of liver, kidney, lung, and knee joint collected from HB mice 4 days after administration of Padua$_{KA}$-HSA$_{QMP}$ or Padua$_{KR}$-HSA$_{QMP}$ (green, anti-FIX; blue, DAPI for nuclei). Images show one representative animal per test article (scale bar: 100 μm). The fluorescence signal at ten different fields per image was quantified. The data represent the group mean ± SD of technical replicates ($n = 10$). Statistical significance was tested by unpaired two-tailed Student's $t$ tests with 95% confidence level (**b–g**, $n = 4$ except $n = 5$ for Padua$_{KR}$-HSA$_{QMP}$, **i-l**, $n = 4$; precise $p$-values are given in Supplementary Tables 6–8; **m** ****$p < 0.0001$ of $n = 10$). FIX-HSA$_{WT}$, black; Padua-HSA$_{QMP}$, gray; Padua$_{KA}$-HSA$_{QMP}$, pink; Padua$_{KR}$-HSA$_{QMP}$, purple. Source data are provided as a Source Data file. **a** Created in BioRender. Hovden Aaen, K. (2025) https://BioRender.com/auh41ln.

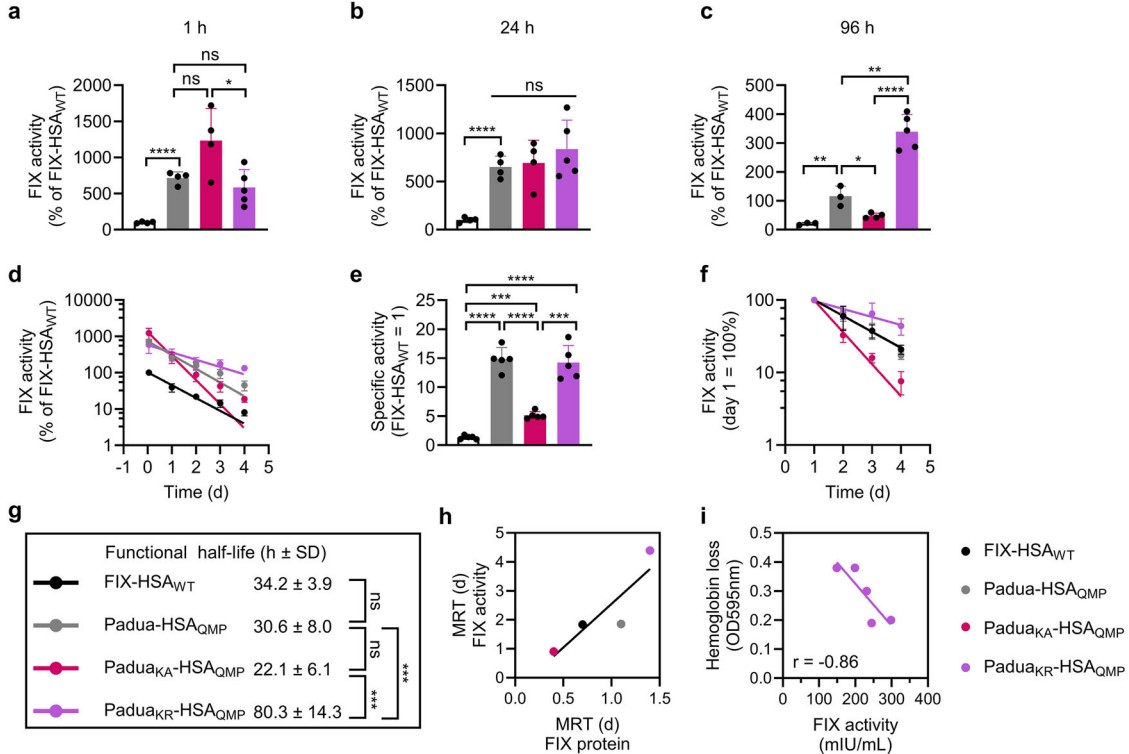

**Fig. 5 | The K5R amino acid substitution grants FIX-HSA enhanced activity and extended functional half-life in HB mice.** FIX activity in plasma of HB mice after (**a**), 1 h, (**b**), 24 h, and (**c**), 96 h post-administration of a FIX-HSA fusion, compared to the activity in plasma of mice that received FIX-HSA$_{WT}$. Data represent the group mean ± SD of biological replicates ($n = 4–5$). **d** Change in FIX clotting activity in plasma over time. Data represent the group mean ± SD of biological replicates ($n = 4–5$) and are presented as the percentage of that of FIX-HSA$_{WT}$ (= 100%). **e** Specific activity measured in mice over time, calculated as the ratio between FIX clotting activity and protein concentrations relative to that of FIX-HSA$_{WT}$ (= 1). Data represent the group mean ± SD at each time point ($n = 5$). **f** Change in FIX activity in plasma over time. Data represent the group mean ± SD of biological replicates ($n = 4–5$) and are presented as the level of clotting activity over time compared to day 1 (= 100%). **g** Functional half-life of the FIX-HSA variants measured by

gPKPDsim. Data represent the group mean ± SD of biological replicates ($n = 4–5$). **h** Correlation between MRT values obtained by applying the FIX plasma concentration (x-axis) and the FIX clotting activity (y-axis) to gPKPDsim. Data represent the group mean ± SD of biological replicates ($n = 4–5$) and were applied to a simple linear regression. **i** Inverse correlation between Padua$_{KR}$-HSA$_{QMP}$ activity and hemoglobin loss upon tail clipping at day 4 post-administration. Data represent the group mean ± SD of biological replicates ($n = 5$) and were applied to a simple linear regression. Statistical significance was tested by unpaired two-tailed Student's $t$ tests with 95% confidence level (**a–d, f–h**, $n = 4$ except $n = 5$ for Padua$_{KR}$-HSA$_{QMP}$; **e** $n = 5$; precise $p$-values are given in Supplementary Tables 9–11). FIX-HSA$_{WT}$, black; Padua-HSA$_{QMP}$, gray; Padua$_{KA}$-HSA$_{QMP}$, pink; Padua$_{KR}$-HSA$_{QMP}$, purple. Source data are provided as a Source Data file.

are limited to 3–4 days, frequent injections are still necessary[7,8,12–14]. Thus, there is a need to further improve their PK properties, as recently achieved by combining the hyperactive FIX Padua (R338L) variant with the engineered HSA$_{QMP}$ variant (Padua-HSA$_{QMP}$)[25]. Motivated by the fact that FIX binds to extravascular Col4[30,47,48], and that this depot alone is sufficient for hemostasis[44,45,50], we engineered Padua-HSA$_{QMP}$ for modulated Col4 binding. When K5A or K5R, which either weakens

or strengthens the ability of FIX to bind Col4[30,31], respectively, were introduced in the FIX-HSA fusion proteins, the FIX clotting activity was reduced or preserved, respectively. Despite this, introduction of the Padua amino acid substitution largely compensated for the reduced activity associated with K5A, as hyperactive properties were observed for all Padua-containing fusion proteins. Importantly, HSA was efficiently cleaved from both the K5A- and K5R-containing fusions upon

**Table 3 | Short- and long-term contributions of K5 engineering in FIX-HSA fusion proteins**

| K5 substitution | Col4 binding | Short-term impact on plasma level (1 h) | Long-term impact (>24 h) | |
|---|---|---|---|---|
| | | | Plasma half-life | Functional half-life |
| K5A | – | ++++ | – | – |
| K5R | ++++ | ++ | ++++ | ++++ |

activation by FXIa, which is required for optimal coagulant activity of FIX[25].

Regarding potential immunogenicity of the engineering strategies used to design the FIX-HSA fusions, we did not detect any warning increase in T- and B-cell epitopes using available prediction tools, as a comparable number of potential epitopes was shown for the FIX variants and the FIX-HSA fusions, including the approved clinical product Idelvion (Supplementary Tables 13 and 14). However, caution should be taken when using such tools, as discussed[65,66].

The favorable FcRn binding properties achieved by QMP engineering of HSA resulted in a 3-fold extended plasma half-life of the FIX-HSA fusion in Tg32 mice. Importantly, whereas the $HSA_{WT}$-containing fusions showed negligible binding to mFcRn, in line with previous reports[54–56,67], the $HSA_{QMP}$-containing fusions bound well to the mouse receptor. Thus, the use of $HSA_{QMP}$ does not only provide a strategy for EHL protein-based therapeutics, but it also partly overcomes the limitations related to the use of mFcRn-expressing mouse models in studies of HSA-based therapeutic candidates. In our study, this allowed us to evaluate the effect of K5 engineering on the PK profile of FIX-HSA in mFcRn-expressing mouse models with different FIX expression profiles, and thus Col4 accessibility, mimicking either CRM+ or CRM− HB patients. In fact, the CRM+ status may modulate the compartment bioavailability of an infused FIX by potential competition with dFIX.

In FIX[plus] Balb/c mice, the presence of an extravascular depot of endogenous FIX mimic the scenario in CRM+ HB patients, including a large fraction of the patients with moderate and severe HB (35% of all with *F9* missense variants)[68,69], which express dFIX that may bind Col4, and thus limit injected FIX from entering this reservoir. Importantly, most missense mutations in the *F9* gene described are not expected to impact the functionality of the Gla domain and its Col4 binding properties[42,69]. In these mice, a negative impact of inefficient Col4 binding was revealed by the shorter half-life measured for $Padua_{KA}$-$HSA_{QMP}$ compared to the other $HSA_{QMP}$-containing variants. On the other hand, K5R engineering of FIX for enhanced Col4 binding might give a competitive advantage over an endogenous dFIX variant expressed at varying levels depending on the individual's *F9* genotype.

Conversely, the *F9* knock-out HB mice mimic the CRM− HB patients with undetectable FIX plasma levels, and thus negligible extravascular depots and freely available Col4, which can capture the injected FIX. One hour after injection, 3-fold more of the K5A-containing variant was detected than of the WT and K5R FIX variants in plasma, supporting that this amino acid substitution results in less capture in the extravascular depot. Interestingly, $Padua_{KA}$-$HSA_{QMP}$ was more rapidly cleared than the other variants, including FIX-$HSA_{WT}$, despite containing $HSA_{QMP}$. In parallel with its rapid clearance, K5A gave a dramatic decline in activity over time. Similar observations have been made for unfused recombinant K5A FIX in the same mouse model[35,50]. In contrast, $Padua_{KR}$-$HSA_{QMP}$ was present at the highest plasma concentration of all variants after 96 hours. In line with its delayed clearance, K5R provided consistently higher clotting activity of $Padua_{KR}$-$HSA_{QMP}$ in plasma over time, resulting in a striking improvement in functional half-life from 30.6 to 80.3 h. Moreover, the K5R substitution resulted in significantly higher concentrations of $Padua_{KR}$-$HSA_{QMP}$ in the liver, kidneys, lungs, and knee joints, with up to a 5-fold difference to that of the K5A-containing fusion. In agreement,

significantly more of $Padua_{KR}$-$HSA_{QMP}$ than $Padua_{KA}$-$HSA_{QMP}$ were detected in the same tissues when stained by IF and IHC. Thus, in the HB mice with freely available Col4, the K5A and K5R substitutions resulted in differential PK profiles and extravascular distribution of the FIX-HSA fusion, due to distinct Col4 binding properties.

Our findings are in line with reports demonstrating that the CRM status affects the bioavailability of infused FIX in mice and humans[42,43,45,46]. In CRM+ patients, depots of Col4-bound dFIX would result in a higher plasma presence of the infused FIX, due to limited Col4 binding. On the other hand, the higher MRT values in FIX[plus] mice expressing mFIX highlight the competitive advantage conferred by the K5R change. Thus, a large body of evidence supports that the extravascular depot of FIX impacts hemostatic competency[41,42,45,70]. Taken together, the experiments performed in the three mouse models revealed the effect of $HSA_{QMP}$ (Tg32 > FIX[plus] ≥ HB) and K5R (HB > FIX[plus] ≥ Tg32) on the PK of the FIX-HSA fusion under different conditions regarding FcRn and FIX expression (Supplementary Table 12). Clearly, the use of $HSA_{QMP}$ as a fusion partner gives a major advantage, as it prolongs the availability of FIX in the circulation. Furthermore, the incremental improvements of the protein engineering steps were evident when correlating the MRT values achieved in the CRM+ (FIX[plus] Balb/c) and CRM− (HB) mouse models, which again highlighted the superior PK properties and competitive advantage of $Padua_{KR}$-$HSA_{QMP}$. Importantly, the decrease in blood loss as a function of $Padua_{KR}$-$HSA_{QMP}$ activity in the HB mouse model indicates a concentration-dependent correction of the disease phenotype, thus underscoring the importance of functional activity and its relevance in therapeutic potential.

In summary, we demonstrate that $Padua$-$HSA_{QMP}$ can be engineered for enhanced extravascular presence by introducing K5R to FIX, resulting in an attractive therapeutic candidate with improved biodistribution and favorable functional profile. Additionally, K5A engineering may provide a tool to improve on-demand treatment with an immediate need of increasing FIX levels in plasma. We provide i) experimental evidence of the impact of Col4 binding as well as CRM status, on the PK and biodistribution of FIX, in the context of HSA, and ii) an in vivo proof-of-concept study demonstrating the translational power of K5 engineering. Our findings support the use of $Padua_{KA}$-$HSA_{QMP}$ and $Padua_{KR}$-$HSA_{QMP}$ as hyperactive short- or long-term therapeutics, with implications for the development of tailorable HB replacement therapies.

## Methods

### Plasmid construction

cDNA constructs encoding WT or Padua (R338L; numbering according to mature protein) FIX (NCBI reference sequences: NM_000133.4, NP_000124.1), the optimized cleavable linker, and mature HSA (amino acids 25-609; NCBI reference sequences: NM_000477.7, NP_000468.1), either $HSA_{WT}$ or $HSA_{QMP}$[24], were produced as described[25]. Codons corresponding to the K5A or K5R substitutions was introduced to cDNA encoding *F9* by site-directed mutagenesis using the QuikChange II XL Site-Directed Mutagenesis Kit (Agilent Technologies, #200521) with the 5′-GTATAATTCAGGTAGATTGGAAGAGTTTG-3′ (K5R) and 5′-GTATAATTCAGGTGCATTGGAAGAGTTTG-3′ (K5A) oligonucleotides (nucleotide changes underlined and reverse primers complementary to the forward ones; numbering of amino acids corresponds to that of mature FIX). cDNA encoding $HSA_{WT}$ was cloned into pFUSE2ss-CLIg-hk vectors (Invitrogen), and site-directed mutagenesis was used to generate the construct encoding $HSA_{QMP}$ (E505/T527M/K573P) (GenScript)[24].

### Protein production and purification

Vectors encoding variants of FIX-HSA fusion proteins, as well as unfused $HSA_{WT}$ and $HSA_{QMP}$, were transfected into Expi293F cells (Gibco, #A14527) using the ExpiFectamine 293 Transfection Kit (Gibco,

#A14525), according to the manufacturer's instructions. For production of FIX-HSA fusion proteins, 100 μg/mL of vitamin K (Konakion Novum; Cheplapharm) was added to the culture prior to transfection. The supernatants were harvested 5 days post-transfection by centrifugation at $(290–400) \times g$ for 25 min in a refrigerated centrifuge and filtered through a 0.2 μm Vacuum Filtration System (VWR). Supernatants containing FIX-HSA proteins were supplied with 1 mM benzamidine hydrochloride hydrate (Sigma-Aldrich, #B6506) to stabilize the proteins.

The supernatant was purified on 5-mL OPUS Chromatography columns pre-packed with a CaptureSelect anti-HSA VHH (Thermo Scientific, #1912970) by Repligen. Before purification, the column was equilibrated with 10 column volumes (CV) of 1X PBS (Sigma-Aldrich, #D8537). The supernatant was loaded onto the column at a flow rate of 0.5–2 mL/min, before the column was washed with 20 CV of PBS. Elution of protein was achieved by running 9 CV of 2 M $MgCl_2$ (Sigma-Aldrich, #63064) at pH 7.4 at a flow rate of 2 mL/min. The eluted protein was buffer exchanged to PBS and further concentrated using an Amicon Ultra-15 centrifugation filter unit (Millipore) with molecular weight cut-offs of 30 kDa or 50 kDa by centrifugation at $(290–400) \times g$ at 4 °C.

Soluble truncated forms of recombinant His-tagged human and mouse FcRn were produced in a Baculovirus expression vector system[71–73]. Briefly, High Five cells (Gibco, #B85502) cultured in Express Five SFM Medium (Gibco, #10486025) were grown to confluency in flat culture flasks at 27 °C, before $1 \times 10^6$ cells/mL were transferred to Erlenmeyer flasks and placed in an orbital shaker set at 160 rpm and further incubated at 27 °C. The cells were transfected with viral stocks of baculovirus encoding a soluble truncated form of either His-tagged mouse or human FcRn, kindly gifted by Dr. Sally Ward (University of Southampton, United Kingdom). After transfection, the cells were cultured for a subsequent 72 h at either 23–24 °C (for hFcRn) or 27–28 °C (for mFcRn) in the orbital shaker. FcRn expressed by the cells was purified from harvested and filtered supernatant by using a 5-mL HisTrap HP column loaded with $Ni^{2+}$ ions (Cytiva, #17524801). Before loading the supernatant (adjusted to pH 7.2), 10 CV of PBS supplemented with 0.05% sodium azide (Sigma-Aldrich, #S2002) was run over the column for pre-equilibration. A maximum of 450 mL of supernatant was run over the column at a flow rate of 5 mL/min. Then, the column was washed with 25 mM imidazole (Sigma-Aldrich, #56748) in PBS (pH 7.2), before elution with 250 mM imidazole in PBS (pH 7.3). Eluted FcRn was further concentrated and the buffer was exchanged to PBS, as described above.

To obtain monomeric protein fractions, size exclusion chromatography (SEC) was performed by using a Superdex 200 Increase 10/300 GL column (Cytiva, #28-9909-44) coupled to an ÄKTA avant 25 instrument (Cytiva) using Unicorn Software (Cytiva), with PBS as running buffer. The monomeric protein fractions were further concentrated using Amicon Ultra-4 Centrifugation Filter Units (Millipore) with molecular weight cut-offs at 30 or 50 kDa by centrifugation at $(290–400) \times g$ at 4 °C, and protein concentration was measured on a DS-11+ spectrophotometer (DeNovix).

## SDS-PAGE
To evaluate the integrity and purity of the protein fractions, a non-reducing sodium dodecyl sulfate polyacrylamide gel electrophoresis (SDS-PAGE) was performed using a Bolt 12% Bis-Tris Plus Gel (Invitrogen, #NW00125). Briefly, 2 μg of protein was mixed with 3 μL of 4X Bolt LDS sample buffer (Invitrogen, #B0007) and the end volume was adjusted to 12 μL with Milli-Q water. The solution was loaded on to the gel in parallel with 7.5 μL of Spectra Multicolor Broad-Range Protein Ladder (Thermo Scientific, #26634) with a range of 10-260 kDa. Electrophoresis was performed at 200 V for 22 min with 1X Bolt MES SDS running buffer (Invitrogen, #B0002) using a PowerPac HC power supply (Bio-Rad Laboratories, Inc.), before the gel was washed 2-3

times in Milli-Q water and stained in Bio-Safe Coomassie Stain (Bio-Rad Laboratories, Inc., #1610786) for 20 min. Next, the staining solution was removed, and the gel was immersed in Milli-Q water overnight before images was acquired using a GelDoc Go Gel Imager (Bio-Rad Laboratories, Inc.). An uncropped SDS-PAGE blot is provided in Supplementary Fig. 18.

## FIX activity assays
The activity of the FIX-HSA fusions was evaluated by an activated partial thromboplastin time (aPTT)-based assay[25] using FIX-deficient plasma (Hyphen BioMed, #ADP050K) and triggering coagulation by the aPTT-S reagent (Sclavo Diagnostic International, #K29005559). Coagulation times were measured on a Coatron X Top coagulometer (TECO Medical Instruments). Chromogenic activity was evaluated using a commercially available kit (Hyphen BioMed, #A221802), following the manufacturer's instructions. Specific activity was calculated as the ratio of activity to protein level (FIX-HSA$_{WT}$ = 1).

## ELISA for evaluation of phospholipid binding
The evaluation of binding between phospholipids and FIX-HSA fusion proteins was performed in an ELISA-based assay, where 96-well plates with U-shaped wells (Thermo Fisher Scientific Nunc) were coated with phospholipids (MP-Reagent; Diagnostica Stago, #86222) diluted 1:800 in carbonate-bicarbonate buffer (pH 9.6) at 4 °C overnight. The wells were blocked with a buffer containing 20 mM Tris, 150 mM NaCl, and 5% (w/v) skimmed milk powder (pH 7.4) for 90 min at room temperature (RT). Next, the wells were washed three times in PBS supplemented with 0.1% (v/v) TWEEN-20 (Sigma-Aldrich, #P1379; 150 μL), before serial dilutions of FIX-HSA variants diluted in sample buffer (20 mM Tris, 150 mM NaCl, 0.5% (w/v) skimmed milk powder, 10 mM $CaCl_2$; pH 7.4) were added to the wells. After incubation for 90 min at 37 °C, the wells were washed as before, and bound proteins were detected by adding a polyclonal goat anti-HSA HRP-conjugated antibody (Bethyl Laboratories, #A80-129P, lot 32) diluted 1:2000 in sample buffer and incubating the wells as before. After washing, o-phenylenediamine diluted in substrate buffer (citrate-phosphate buffer; pH 5.0) was added to the wells. The colorimetric reaction was stopped by addition of 2.5 M $H_2SO_4$. Absorbance was read using a Sunrise Absorbance Microplate Reader (TECAN).

## Time-course evaluation of HSA detachment from FIX-HSA upon cleavage by FXIa
The detachment of HSA from the FIX-HSA fusion proteins upon cleavage by activated FXI (FXIa) was evaluated by time-course incubation of purified fusion variants with plasma-derived FXIa (pdFXIa; Haematologic Technologies, #HCXIA-0160) diluted in reaction buffer (20 mM HEPES, 150 mM NaCl, 0.1% (v/v) PEG-8000, 5 mM $CaCl_2$) at pH 7.4[74]. Western blotting was performed using polyclonal goat anti-human albumin (1:1000 dilution) and polyclonal donkey anti-goat IgG HRP-conjugated (1:4000 dilution) antibodies (Bethyl Laboratories, #A80-129A, lot 10; Bethyl Laboratories, #A50-101P, lot 23) in PBS supplemented with 2.5% (w/v) skimmed milk powder (Sigma-Aldrich, #70166). Blotting images were acquired using a ChemiDoc instrument (Bio-Rad Laboratories, Inc.) and analyzed by Image Laboratory (version 4.0; Bio-Rad Laboratories, Inc.). Uncropped Western blots are provided in Supplementary Fig. 19.

## ELISA for evaluation of FcRn binding
ELISA-based FcRn binding studies were performed by coating 96-well EIA/RIA Clear Flat Bottom Polystyrene Microplates (Corning) with 8 μg/mL of a human IgG1 with the M252Y/S254T/T256E/H433K/N434F (MST/HN)[75] substitutions in the Fc region that has specificity for 4-hydroxy-3-iodo-5-nitrophenylacetic acid (NIP)[72] overnight at 4 °C (100 μL/well), followed by blocking with

PBS containing 4% (w/v) skimmed milk (PanReac AppliChem, #A0830; PBSM) at RT for 1 h on tilting. In all following steps, including the washing steps, buffers at either pH 5.5 or pH 7.4 were used. The PBS used for the pH 5.5 buffer was composed of 177 mM phosphate and 85 mM NaCl, while that for the pH 7.4 buffer was composed of 195 mM phosphate and 85 mM NaCl. Next, 10 µg/mL of monomeric truncated forms of soluble his-tagged hFcRn or mFcRn in PBSM with 0.005% TWEEN-20 (PBSTM) was added to all wells (100 µL/well) and incubated at RT with tilting for 1 h, before the wells were washed five times in 200 µL PBST. Next, serial dilutions of FIX-HSA variants (0–75 nM, 1:1 dilution in 7 steps) in 100 µL PBSTM were added. After subsequent incubation for 1 h at RT, the wells were washed as prior, before detection was performed by adding 125 ng/mL of an alkaline phosphatase (ALP)-conjugated goat polyclonal anti-HSA antibody (Bethyl Laboratories, Inc., #A80-229AP, lot 10) in PBSTM at 100 µL/well. Lastly, 1 mg/mL p-nitrophenyl phosphatase substrate (Sigma-Aldrich, #S0942) diluted in Pierce Diethanolamine Substrate Buffer (pH 9.8; Thermo Scientific, #34064) was added (100 µL/well) and absorbance was read at 405 nm using a Sunrise Absorbance Microplate Reader (TECAN).

## Surface plasmon resonance (SPR)

Binding kinetics were determined by SPR using a Biacore T200 instrument (Cytiva). FIX-HSA fusion proteins were immobilized (200 RU) on CM5 Series S sensor chips (Cytiva, #29149603) in 10 mM sodium acetate at pH 4.5 (Cytiva, #BR100350) using an amine coupling kit (Cytiva, #BR100050). Next, the chip was washed with 50 mM NaOH and unreacted moieties were blocked with 1 M ethanolamine. Thereafter, serial dilutions of monomeric truncated versions of soluble his-tagged hFcRn (15.6–2000.0 nM for $HSA_{WT}$, 1.95–62.5 nM for $HSA_{QMP}$) or mFcRn (7.8–8000.0 nM for $HSA_{WT}$, 3.9–500 nM for $HSA_{QMP}$) were injected (temperature: 25 °C, injection time: 120 s, dissociation: 680–2000 s, regeneration: 60–120 s, flow rate: 40 µL/min). The running buffer was a phosphate buffer containing 0.005% (v/v) TWEEN-20 at either pH 5.5 (177 mM phosphate, 85 mM NaCl) or pH 7.4 (195 mM phosphate, 85 mM NaCl). Regeneration was performed with the pH 7.4 phosphate buffer. Binding curves were zero adjusted and the reference flow cell value was subtracted. The 1:1 Langmuir binding model provided by the Biacore T200 Evaluation Software version 3.0 (Cytiva) was used to determine kinetic rate constants and affinity values.

## Human endothelial cell-based recycling assay (HERA)

HERA[57] was performed to evaluate FcRn-mediated cellular handling of the FIX-HSA fusion proteins. Briefly, $1.5 \times 10^5$ cells/mL HMEC-1 cells stably expressing hFcRn-EGFP (HMEC-1-hFcRn)[76], gifted by Dr. Wayne I. Lencer (Boston Children's Hospital, Harvard Medical School and Harvard Digestive Diseases Center, USA), were seeded into 48-well cell culture plates (Costar) in 250 µL/well of growth medium consisting of MCDB-131 medium (Gibco, #10372019) supplemented with 10% (v/v) heat-inactivated fetal bovine serum (FBS; Gibco, #F7524), 25 U/mL penicillin and 25 µg/mL streptomycin (Gibco, #P4458), 2 mM L-glutamine (Gibco, #25030081), 1 µg/mL hydrocortisone (Sigma-Aldrich, #H0888), and 10 ng/mL recombinant mouse epidermal growth factor (Gibco, #PMG8041). To maintain a high expression of hFcRn, 5 µg/mL Blasticidine S HCl (Gibco, #A1113903) and 250 µg/mL Geneticine Selective Antibiotic (G418 Sulfate; Gibco, #10131035) were added to the growth medium.

The cells were incubated until confluent (approximately 24 h after seeding) at 37 °C and 5% $CO_2$, before the medium was removed. Each well was washed 3 times in 200 µL of pre-heated Hank's buffered saline solution (HBSS; Gibco, #14025100). Next, the cells were "starved" in pre-heated HBSS for 1 h at 37 °C, before 800 nM of HSA variants, either fused to FIX or unfused, diluted in 125 µL of RT HBSS (pH 7.4) were added to the wells. To allow for cellular uptake, the cells were

incubated for 3 h at 37 °C and 5% $CO_2$. Next, the cells were washed five times in 200 µL of ice-cold HBSS, before the plate was stored at −80 °C until the day of analysis. A parallel plate was treated identically, and washed as described, before 220 µL of recycling medium was added to each well. The recycling medium consisted of complete growth medium without FBS, G418 sulfate, or Blastidine S HCl, supplemented with 1X Eagle's MEM non-essential amino acids solution (Gibco, #11140050). To allow for ligand recycling, the cells were incubated for 3 h at 37 °C and 5% $CO_2$. Next, the cell media were collected and stored at −20 °C until the day of analysis.

At the day of analysis, cell lysates were prepared from the plate stored at −80 °C by adding 220 µL of Pierce RIPA Lysis and Extraction Buffer (Thermo Scientific, #89900) supplemented with 1X cOmplete protease inhibitor cocktail (Roche, #11836145001) to each well and incubating the plate for 10 min on ice with tilting. Directly after, the plate was centrifuged at $(290–400) \times g$ at 4 °C for 10 min, before cell lysates were collected and kept on ice. The medium samples were thawed on ice. The cell lysate and medium samples were analyzed by a two-way anti-HSA ELISA.

## ELISA for protein quantification in HERA and plasma samples

A two-way anti-HSA ELISA was used to quantify the FIX-HSA fusion proteins and unfused HSA present in the uptake and recycling samples from HERA as well as in plasma samples from Tg32 and FIX[plus] (Balb/c) mice. Briefly, 96-well EIA/RIA Clear Flat Bottom Polystyrene Microplates (Corning) were coated with either 8 µg/mL of polyclonal goat anti-HSA antibody (Sigma-Aldrich, #A1151, lot SLCF5233; for analysis of HERA samples) or 1 µg/mL of monoclonal goat anti-HSA antibody (15C7; Abcam, #MA1-90420, lot YE3901934; for analysis of plasma) diluted in PBS at 4 °C overnight (100 µL/well) and subsequently blocked with PBSM for 1 h at RT with tilting. Next, a dilution series of each sample (HERA: 1:1, 1:2, 1:4 dilutions; plasma: 1:50, 1:100, 1:200 dilutions) in PBSTM was added alongside a concentration gradient of each respective FIX-HSA variant (HERA: 0–4 nM, plasma: 0–8 nM; 1:1 dilution in 11 steps) in PBSTM (100 µL/well). After 1 h of incubation at RT with tilting, the plate was washed 3 times in 200 µL of PBST. Protein was detected by adding 125 ng/mL of an ALP-conjugated polyclonal goat anti-HSA antibody (Bethyl Laboratories, Inc., #A80-229AP, lot 10) diluted in PBSTM (100 µL/well). Lastly, 1 mg/mL of p-nitrophenyl phosphatase substrate diluted in diethanolamine buffer (pH 9.8) was added (100 µL/well) and absorbance was read at 405 nm using a Sunrise Absorbance Microplate Reader (TECAN).

An anti-FIX/anti-HSA ELISA was used to quantify protein in plasma from the study in HB mice, where wells were coated with 2.5 µg/mL polyclonal goat anti-FIX antibody (Cedarlane Laboratories, #CL20040A, lot AIG2241-1R6) before blocking with PBSM (100 µL/well). The standard was a pure fraction of the Padua-$HSA_{QMP}$ fusion protein added in a concentration gradient (0–16 nM, 1:1 dilution in 11 steps), and the plasma samples were added as 1:10–1:160 (1:1 diluted in each step), both in PBSTM. After 1 h at RT on a tilting board, the plates were washed 3 times in 200 µL of PBST, before the proteins were detected as described above. FIX in tissue homogenates from HB mice was quantified by a polyclonal anti-human FIX ELISA kit (Affinity Biologicals, #FIX-EIA, lot EIA9-0058R1), following the manufacturer's instructions[77].

## ELISA for quantification of mFIX

To quantify the expression level of endogenous mFIX in Tg32, FIX[plus] (Balb/c) and HB mice, plasma samples from these mouse models were analyzed using an mFIX-specific ELISA kit (Novus Biologicals, #NBP2-82408, lot XF19T62R9186), following the manufacturer's instructions. Briefly, 100 µL of an mFIX reference solution (0.16–10 ng/mL) was added to wells pre-coated with an anti-FIX antibody, in duplicate. In parallel, 100 µL of plasma samples were added to wells in a 1:200 dilution in the "reference & standard solution buffer", in duplicates. After incubation at 37 °C for 90 min, the reference and standard

solutions were removed from the wells, and 100 μL of working strength detection antibody was added to each well. The plate was incubated for 1 h at 37 °C, before the wells were washed 3 times with 350 μL of working strength wash buffer. During each washing step, the buffer was left in the wells for 1–2 min before being removed by decanting. Immediately after, 100 μL of working strength HRP conjugate was added to each well, and the plate was incubated at 37 °C for 30 min. Next, the wells were washed with the same procedure as before, however, five times. Lastly, 90 μL of substrate solution was added to each well and the plate was incubated for 25 min at RT before the reaction was stopped with 50 μL of stop solution. Absorbance was read at 450 nm using a Sunrise Absorbance Microplate Reader (TECAN) after the signal had stabilized.

### Animal studies

All animals of the study were housed in a controlled environment, with a 12-h light/dark cycle, ventilated cages, regulated ambient temperatures ($21 \pm 2$ °C at Oslo University Hospital, Norway; $22 \pm 2$ °C at IRCCS San Raffaele, Italy) and relative humidity (30–70% at Oslo University Hospital, Norway; 50–60% at IRCCS San Raffaele, Italy). Bedding and nesting material were changed regularly, and food was accessed ad libitum. The mice were euthanized by $CO_2$ or cervical dislocation at termination of the experiments.

### PK study in Tg32 mice

A study to examine the plasma half-life of FIX-HSA fusions in homozygote hFcRn transgenic Tg32 mice (B6.Cg-Fcgrttm1Dcr Tg(FCGRT)32Dcr/DcrJ)[58] was performed at Oslo University Hospital (KPM, Rikshospitalet, Oslo, Norway) with approval from the Norwegian Food Safety Authority (national authorization number: 23998), using animals obtained from the Jackson Laboratory (Bar Harbor, ME, USA). Each mouse (male/female, 16 weeks, 20–30 g, 5 mice/group) received 2 mg/kg of FIX-HSA in PBS by I.V. injection. Blood (25 μL) was collected from the saphenous vein at 1, 1.5, 2, 3, 4, 5, 9, and 11 days post-injection and processed to plasma before storage at −20 °C until the day of analysis. The samples were analyzed by ELISA and remaining fusion protein in plasma was reported as protein concentration in plasma at each time point, percentage remaining compared to the concentration 1 day post-administration (day 1 = 100%), and as concentration (μg/ml) at 24 h. The β-phase plasma half-life was calculated using the following formula; $t_{1/2\beta} = \log 0.5/(\log Ae/A0) \times t$ ($t_{1/2\beta}$: β-phase half-life, Ae: concentration remaining, A0: concentration on day 1, t: elapsed time).

### PK study in Balb/c (FIX[plus]) mice

A study to examine plasma the half-life of FIX-HSA fusions in WT mice (Balb/c, FIX[plus] mice; BALB/cAnNRj, strain #0003, Janvier Labs) was performed at Oslo University Hospital (KPM, Rikshospitalet, Oslo, Norway) with approval from the Norwegian Food Safety Authority (national authorization number: 26082). The mice (male, 12–13 weeks, 20–30 g, 5 mice/group) received 2.5 mg/kg of FIX-HSA diluted in PBS by I.V. injection. Blood (0.7 μL/g) was collected from the saphenous vein at 6, 12, 27, 50, 74, and 96 h (the last time point applied to HSA_QMP variants only) post-administration and processed to plasma by centrifugation before storage at −20 °C until the day of analysis. The samples were analyzed by ELISA and the remaining fusion protein in plasma and β-phase plasma half-life was reported as described for the PK study in Tg32 mice.

### PK and distribution study in HB mice

A study to examine the PK profile and biodistribution of FIX-HSA proteins in *F9* KO mice (C57BL/6J-F9[tm1Dws]; HB mice)[60] was performed at the IRCCS San Raffaele (Milan, Italy) with approval from the Institutional Animal Care and Use Committee (authorization numbers: 1141

and 1329). The mice (male/female, 8–10 weeks, 30 g, 4–5 mice/group) were administered 2.5 mg/kg of FIX-HSA fusion protein diluted in PBS by I.V. injection. Blood (100 μL) was sampled from the retro-orbital plexus by using capillary tubes after 1 h and thereafter every day up to 4 days (1–4 days). Blood was collected into 0.38% sodium citrate buffer (pH 7.4) and processed to plasma by centrifugation at $3300 \times g$ for 5 min before immediate storage at −80 °C. Plasma samples from all mice 72 h pre-treatment were collected as reference samples. The samples were analyzed by ELISA and the remaining fusion protein in plasma and β-phase plasma half-life was reported as described for the PK study in Tg32 mice. FIX activity in plasma from HB mice was analyzed by aPTT-based assays[25,77], with the activity of pooled normal plasma (PNP) as reference and plasma from untreated mice as baseline.

### Tail clip bleeding assay on HB mice

To assess hemostasis in mice receiving a FIX-HSA fusion, a tail clip bleeding assay was performed in HB mice[78]. The mice ($n = 5$) were anesthetized 4 days post-administration of Padua_KR-HSA_QMP and their tails were placed in pre-warmed water at 37 °C for 2 min and subsequently cut at 2.5–3 mm diameter. Immediately after, the tails were placed in PBS supplemented with calcium and magnesium at 37 °C. Bleeding or clotting was monitored for 15 min. Erythrocytes were collected by centrifuging the blood-containing PBS at $520 \times g$ for 10 min at 4 °C and resuspending in 6 mL lysis buffer (10 mM $KHCO_3$, 150 mM $NH_4Cl$, 0.1 mM EDTA). After incubation for 10 min at RT, the samples were centrifuged as before, and absorbance was read at 595 nm.

### Homogenization and processing of organ samples from HB mice

At termination of the study in HB mice (day 4), liver, kidneys, lungs, and knee joints were collected from each mouse and immediately snap frozen at −80 °C. To improve sample processing, homogenization was performed on ice using the T-10 basic ULTRA-TURRAX homogenizer (IKA-WERKL, Germany) with frozen organs added to Tris-buffered saline containing 10% (w/w) sodium citrate. Homogenates were centrifuged at 4 °C at $1750 \times g$ for 15 min. The primary supernatant was collected and transferred to a clean tube before a second centrifugation, performed as above. Immediately after, the resulting supernatant was collected and moved to a clean tube before storage at −80 °C. For IF and IHC analysis, harvested livers, lungs, kidneys, and knee joints were fixed for 24 h in zinc-formalin. Embedding of organs and joints were performed by the Center of Experimental Imaging facility at San Raffaele Hospital (Milan, Italy). Briefly, formalin-fixed paraffin-embedded consecutive sections (4 μm) were dewaxed and hydrated through series of decreasing alcohol gradients.

### IF staining of tissues from HB mice

Sections (4 μm) of paraffin-embedded organs harvested from HB mice were cut by microtome (Leica Biosystems), deparaffinized in xylene, rehydrated in a decreasing alcohol gradient (100%, 90%, 80%, 70%, and 30%; v/v) and lastly washed three times with deionized water. The sections were then heated in sodium citrate buffer (pH 6.0) (Merck Millipore) for antigen retrieval at a sub-boiling temperature of 95 °C in a water bath for 10 min. The samples were placed on ice and cooled to RT for 25 min, washed twice with PBS and then blocked with PBS supplemented with 5% (w/v) bovine serum albumin (BSA; Sigma-Aldrich, #A2153) for 45 min at RT. Sections were then incubated overnight at 4 °C with the following primary antibodies; polyclonal rabbit anti-collagen IV antibody (abcam, #ab6586, lot 1084886-5) or polyclonal goat anti-human FIX antibody (Affinity Biologicals, #GAFIX-AP, lot AP4007-DR1), diluted 1:200 in PBS supplemented with 0.25% (w/v) BSA. The day after, the samples were washed three times with PBS and detection was performed with an Alexa Fluor 488-conjugated goat anti-rabbit or donkey anti-goat antibodies (Thermo Fisher Scientific, #A-11034, lot 2380000 and #11055, lot 2411589), diluted 1:500 in PBS supplemented with 0.25% (w/v) BSA for 45 min at RT. Nuclei

were stained with DAPI (Thermo Fisher Scientific, #D1306). Coverslips were mounted onto glass slides using Epredia Lab Vision PermaFluor aqueous mounting medium (Thermo Fisher Scientific, #TA-030-FM), and images were acquired using the Nikon Ci-L microscope connected to a DS-Qi2Mc digital camera equipped with the NIS-ELEMENTS software (Nikon; version D). Finally, the fluorescence signal in the images was quantified using ImageJ software 1.53a (Java 1.8.0_172, National Institutes of Health).

## IHC analysis of tissues from HB mice

Immunohistochemical sections of tissues from HB mice were deparaffinized, rehydrated in distilled water through an ethanol gradient (100%, 90%, and 70%; v/v), before processing with 3% (v/v) $H_2O_2$ in PBS for 10 min and blocking with 2.5% (v/v) normal horse serum (Vector-Labs, #MP-7405) for 30 min at RT. The slides were incubated overnight with polyclonal goat anti-human FIX or polyclonal rabbit anti-collagen IV primary antibodies (Affinity Biologicals, #GAFIX-AP, lot AP4007-DR1; Abcam, #ab6586, lot 1084886-5) diluted 1:100 in PBS supplemented with 0.25% (w/v) BSA. The day after, the slides were washed with PBS for 5 min before incubation with ImmPRESS (peroxidase) polymer anti-rabbit IgG reagent (Vectorlabs, #MP-7401, lot WOVUS27503) or ImmPRESS (peroxidase) polymer anti-goat IgG reagent (Vectorlabs, #MP7405, lot ZL0829), respectively, for 30 min. After washing twice in PBS for 5 min, the sections were incubated with a 3,3'-diaminobenzidine solution for 2 min, washed in PBS for 3 min, then cleared and mounted in glycerol. Images were acquired using a Nikon Microphot FXA brightfield microscope.

## PK and pharmacodynamic (PD) analysis

The protein concentrations in plasma from all animal studies were applied to the gPKPDsim application[59,79] using MATLAB (Version R2020b) for prediction of PK and pharmacodynamic (PD) values. Functional half-life was calculated by applying the FIX activity levels measured in plasma to gPKPDsim.

## T- and B-cell peptide epitope prediction

Potential T-cell epitopes were predicted by using the NetMHC4.1 tool, with 9-mer peptides against representative human HLA supertypes[80]. The default rank thresholds for strong (0.5) and weak (2.0) binders were used. Potential B-cell peptide epitopes were predicted using the Immune Epitope Database (IEDB) Bepipred Linear Epitope Prediction (BepiPred) 2.0, with a threshold of 0.5[81].

## Statistical analysis

Statistical differences were analyzed by two-tailed unpaired Student's $t$ tests in GraphPad Prism (Version 9.3.1) with $p < 0.05$ defined as statistically significant and a 95% confidence level.

## Reporting summary

Further information on research design is available in the Nature Portfolio Reporting Summary linked to this article.

## Data availability

Data supporting the findings of this study is available in the article, its Supplementary information, the Source Data file and from the corresponding authors upon request. Source data are provided with this paper.

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

## Acknowledgements

K.H.A. was supported by the Research Council of Norway (Grant no. 287927). J.N., S.B. and J.T.A. were supported by the Research Council of Norway (Grant no. 274993). J.T.A. was supported by the South-Eastern Norway Regional Health Authority (Grant no. 2018052; 2019084; 2024046). M.L.H. was supported by the Norwegian Cancer Society (Grant no. 223315). This work was partially supported by the Research Council of Norway through its Centres of Excellence scheme (project number 332727). A.B., M.F.T., R.T., F. B., and M.P. were supported by the Access to Insight Basic Research Grant (Novo Nordisk) and by the Fondo di Ateneo per la Ricerca (FAR) from the University of Ferrara, Italy.

## Author contributions

K.H.A., A.B., and J.T.A. designed the research. K.H.A., M.F.T., J.N., R.T., A.C., C.C., M.B., S.B., and A.B. performed the research. M.N.A., M.L.H., and G.V. contributed with reagents. K.H.A., M.F.T., M.B., F.B., M.P., A.B., and J.T.A. analyzed the data. K.H.A., J.N., I.S., F.B., M.P., A.B., and J.T.A. wrote the paper. All authors reviewed and approved the final version of the manuscript.

## Competing interests

J.T.A. and I.S. are co-inventors of patents, which are entitled "Albumin Variants and uses thereof" (e.g. EP3063171B1, US10208102 and US10781245) and relate to the data described in this paper. The remaining authors declare no conflict of interest.
