## [Transparent Peer Review file · Nature Communications]

Tailored collagen binding of albumin-fused hyperactive coagulation factor IX dictates in vivo distribution and functional properties

Corresponding Author: Professor Jan Terje Andersen

Version 0:

Reviewer comments:

Reviewer #1

(Remarks to the Author)

Kristin et al. demonstrated different strategies for changing activity or half-life of FIX protein using a combination of hyperactive mutant FIX (Padua), enhanced or impaired affinity to Col4: K5R and K5A mutation, and engineered HSA which displayed enhanced affinity to FcRn. These strategies can dramatically impact the pharmacokinetics of FIX protein. This study also showed the effect of binding ability to Col4 with K5 mutations in vivo. The increasing of extravascular distribution of K5R mutation can enhance the functional half-life. The K5A mutation also showed the highest FIX level at early time points and can be used for on-demand treatment with an immediate need of FIX levels in plasma. Additionally, the authors also provide the half-life data of EHL FIX protein in Tg32 transgenic mice, FIXplus mice and HB mice to mimic the CRM+ or CRM- HB patients. In summary, the combination of the hyperactive mutation (R338L) in FIXPadua, enhanced or decreased binding to Col4 (K5R and K5A, respectively), and improved engagement with HAS (HSAQMP) can significantly enhance the efficacy of replacement therapy for HB for different purposes. This new EHL FIX protein (PaduaKA-HSAQMP and PaduaKR-HSAQMP) can be used in either short- or long-term therapeutics according to the requirement.

The data overall are well presented, and the paper is generally well written. Notably, FIX Padua is recognized as a hyperactive FIX mutation, and HSAQMP enhances FcRn binding affinity, substantially improving the half-life of FIX, as demonstrated in Tg32 mice, which possess hFcRn to mimic the human condition. One question is why FIX K5 engineering has a lesser effect on FIX MRT when comparing K5 (WT) and K5R (K5 vs. K5R) in HB mice (1.2 vs. 1.4) than in FIXplus mice (1.4 vs. 1.9). Considering that HB mice are CRM- phenotype and their extracellular depots are not saturated by non-functional FIX protein, the discrepancy in MRT between K5 and K5R should be more pronounced than in FIXplus mice. If the varying binding affinity between human and mouse collagen IV to human FIX is speculated as one of the reasons or the authors have different perspective, those points should be addressed in the discussion section.

Major issues:

1. Why did FIX with the K5A mutation (PaduaKA-HSAQMP or FIXKA-HSAWT) exhibit lower clotting activity (Table S1) and specific activity compared to K5 (Padua-HSAQMP or FIX-HSAWT) and K5R (PaduaKR-HSAQMP or FIXKR-HSAWT)? (Figure 1d). Did the KA mutation affect FIX activity?
2. In Tg32 mice, HSAQMP exhibited significantly higher MRT compared to HSAWT, attributable to the hFcRn transgene. However, this difference was not observed in either FIXplus or HB mice. Moreover, in FIXplus and HB mice, the increase in MRT from Padua-HSAQMP to PaduaKR-HSAQMP was marginal, while the MRT decreased further when comparing FIX-HSAWT and FIXKR-HSAWT in Tg32 mice (Table 2). Is there evidence suggesting that this phenomenon is linked to the binding affinity between mouse collagen IV and human FIX? Would the use of human collagen IV potentially amplify the difference between Padua-HSAQMP and PaduaKR-HSAQMP?
3. Why does PaduaKA-HSAQMP exhibit lower MRT compared to FIX-HSAWT in HB mice but not in FIX plus mice? Could the presence of CRM+ impact the clearance of PaduaKA-HSAQMP?

4. The specific activity of FIX is defined as the units of factor IX per milligram. It is recommended that FVIII activity (IU) levels are used instead of clotting time in figure 1b and 1c to avoid confusion.

5. Will any of these mutations or their combination have any impact on the immunogenicity of FIX protein?

Minor issues:

1. Figure 4f cannot see the % remaining of PaduaKR-HSAQMP in the figure. Is that because of overlapping with another group? Maybe using different symbols will help make the figure clearer.

2. Figure 5f cannot see Padua-HSAQMP group clearly. Is that due to overlap with FIX-HSAWT? Again, maybe using different symbols will help make the figure clearer.

Reviewer #2

(Remarks to the Author)

Reviewer #3

(Remarks to the Author)

The authors investigated the effect of different mutations of FIX on its pharmacokinetic and pharmacodynamic effects. More in details it has been investigated the impact of engineering FIX for improved (K5R) or reduced (K5A) Col4 binding, on the pharmacokinetic 48 (PK) profile of hyperactive FIX Padua fused to an engineered human serum albumin (HSA; 49 HSAQMP) exhibiting favorable neonatal Fc receptor (FcRn) engagement. The results clearly demonstrated that PaduaKA-HSAQMP showed a shorter half lifetime and activity compared to PaduaKR-HSAQMP which has a much longer effect. This study paves the way for future therapeutic opportunities for optimization and personalization of HB replacement therapy.

Major comments

1) The authors should implement the data regarding the tissue accumulation of that PaduaKA-HSAQMP compared to PaduaKR-HSAQMP. They should perform IHC analysis from lungs, kidneys, liver, and knee joints tissue in order to show the localization of FIX compared to collagen IV. This analysis is essential to confirm the general hypothesis on the different binding of PaduaKR-HSAQMP and PaduaKA-HSAQMP to collagen.

2) Please provide the protocol number for authorization of animal studies

Reviewer #4

(Remarks to the Author)

The authors have reported on the in vivo distribution and functional characteristic of increased or decreased collagen binding potentials of albumin-fused gain-of-function hyperactive FIX (FIX-Padua) compared to WT. As the authors have also described, we also know that the efficacy of HB replacement therapy is evaluated based on the FIX:C levels in blood circulation plasma, while FIX which binds to extravascular type IV collagen appears to contribute to possible hemostasis and to affect the biodistribution of infused rFIX. In the present study, they investigated the impact of engineering FIX for increased (K5R substitution) or decreased (K5A substitution) type IV collagen binding potential mutant, on the PK profile of FIX-Padua fused to an engineered human albumin (HSAQMP).

Their obtained results can be summarized as below;

1: Regardless of the K5 mutation, Padua-HSA, PaduaK5R-HSA, and PaduaK5A-HSA showed hyperactive properties and improved hFcRn binding and cellular recycling, resulting in extended half-life in hFcRn-transduced mice.

2. In mice mimicking CRM-positive patients expressing dysfunctional FIX, K5A-product negatively modulated the plasma half-life. In CRM-negative HB mice, K5A and K5R product exerted opposite effects on the PK and biodistribution profiles of the fusion.

Therefore, PaduaK5A-HSA showed no extravascular distribution, and the highest plasma levels at early time points. While,

PaduaK5R-HSAQMP showed increased extravascular distribution to the center of liver organ, and knee joints, and 3-fold longer functional half-life.

The authors concluded that the impact of collagen binding potential on the PK profile of FIX fused to HSA, supporting the use of PaduaK5A-HSA and PaduaK5R-HSA as hyperactive short- or long-term therapeutics with opportunities for personalization of HB replacement therapy.

The concept and methodology of the present study appears to be reasonable and the obtained results may be enough and clear to explain the authors final conclusion. The development of tailored-made or personalized FIX product may be hoped. However, some major concerns can be raised in the present study.

1. Did the authors perform these biochemical or enzymatic analyses on the FIX-modified gain- or loss-of-function proteins generated in the present study? Did albumin-fused proteins completely albumin portion release and function as well as FIXa, when converted to FIXa? How about the properties of FIX/FIXa? Furthermore, did you investigate the properties of the intrinsic tease complex in vitro (association with FX or FVIIIa), binding potential to phospholipid membrane, and reaction with FXIa or FVIIa etc.? These results would be necessary because this is stated as a future therapeutic application that the authors conclude.
2. In the present study, FVIII:C was measured by the one-step method, but was it evaluated by a chromogenic assay?
3. Related to the above comment, since this is a mouse-based experiment, all coagulation factors are mouse-based coagulation proteins, and since FIX is a human FIX mutant protein, the differences between mouse-derived coagulation factor reactions and human-derived coagulation protein reactions should be fully considered.
4. Although the current study was evaluated with FIX activity measurement, real data on global coagulation potential and hemostatic potential are also needed to lead to your conclusions from this study. Data on comprehensive coagulation measurement using for example, ROTEM/TEG and thrombin generation, and hemostatic potential measurement such as tail-cut bleeding should also be evaluated.
5. The present study mentioned the focus point of CRM+ in terms of extravascular depot of endogenous FIX. However, the mechanism induced CRM+ pattern is not only about this mechanism. It is not appropriate to discuss CRM+ as a whole only in terms of extravascular depot of endogenous FIX. It seems too much to say that it is indicated for treatment.
6. What is the immunogenicity of the FIX modified proteins which were generated this time? If it is supposed to be applied to hemostatic therapy, I think this analysis should be fully analyzed. for example, in silico analysis etc.

Version 1:

Reviewer comments:

Reviewer #1

(Remarks to the Author)

The authors have adequately addressed all the concerns.

Reviewer #2

(Remarks to the Author)

Reviewer #3

(Remarks to the Author)

I do not have any further comments

Reviewer #4

(Remarks to the Author)

Thank you very much for your full response to my many questions and comments.

I carefully have read the response to reviewers and revised manuscript. The authors have addressed the full responses using additional experiments and comments for all questions. I agree with your comments. I think that this paper is worthy of publication in this journal.

Ref: Ms. No. NCOMMS-24-13282A

Aaen-Testa et al.

Title: Tailored collagen binding of albumin-fused hyperactive coagulation factor IX dictates in vivo distribution and functional properties

POINT TO POINT RESPONSE TO REVIEWERS

REVIEWER #1

Kristin et al. demonstrated different strategies for changing activity or half-life of FIX protein using a combination of hyperactive mutant FIX (Padua), enhanced or impaired affinity to Col4: K5R and K5A mutation, and engineered HSA which displayed enhanced affinity to FcRn. These strategies can dramatically impact the pharmacokinetics of FIX protein. This study also showed the effect of binding ability to Col4 with K5 mutations in vivo. The increasing of extravascular distribution of K5R mutation can enhance the functional half-life. The K5A mutation also showed the highest FIX level at early time points and can be used for on-demand treatment with an immediate need of FIX levels in plasma. Additionally, the authors also provide the half-life data of EHL FIX protein in Tg32 transgenic mice, FIXplus mice and HB mice to mimic the CRM+ or CRM- HB patients. In summary, the combination of the hyperactive mutation (R338L) in FIX Padua, enhanced or decreased binding to Col4 (K5R and K5A, respectively), and improved engagement with HSA (HSA_{QMP}) can significantly enhance the efficacy of replacement therapy for HB for different purposes. This new EHL FIX protein (Padua_{KA}-HSA_{QMP} and Padua_{KR}-HSA_{QMP}) can be used in either short- or long-term therapeutics according to the requirement.

The data overall are well presented, and the paper is generally well written. Notably, FIX Padua is recognized as a hyperactive FIX mutation, and HSA_{QMP} enhances FcRn binding affinity, substantially improving the half-life of FIX, as demonstrated in Tg32 mice, which possess hFcRn to mimic the human condition. One question is why FIX K5 engineering has a lesser effect on FIX MRT when comparing K5 (WT) and K5R (K5 vs. K5R) in HB mice (1.2 vs. 1.4) than in FIXplus mice (1.4 vs. 1.9). Considering that HB mice are CRM- phenotype and their extracellular depots are not saturated by non-functional FIX protein, the discrepancy in MRT between K5 and K5R should be more pronounced than in FIXplus mice. If the varying binding affinity between human and mouse collagen IV to human FIX is speculated as one of the reasons or the authors have different perspective, those points should be addressed in the discussion section.

Major issues:

1. Why did FIX with the K5A mutation (Padua_{KA}-HSA_{QMP} or FIX_{KA}-HSA_{WT}) exhibit lower clotting activity (Table S1) and specific activity compared to K5 (Padua-HSA_{QMP} or FIX-HSA_{WT}) and K5R (Padua_{KR}-HSA_{QMP} or FIX_{KR}-HSA_{WT})? (Figure 1d). Did the KA mutation affect FIX activity?

We thank the reviewer for raising this point and we do agree that our data point towards a reduced specific activity for the KA variant as compared with the wild-type (WT) or KR-bearing variant.

To provide additional evidence, we now report data on antigen and activity levels, as well as specific activity, of unfused FIX with the KR or KA substitution. The results show that the activity of unfused KR is similar to that of WT FIX (93 ± 10% of WT FIX; specific activity, 0.98 ± 0.1),

while unfused KA shows decreased activity ($54 \pm 10\%$ of WT FIX; specific activity, 0.5 ± 0.05). This is in line with the reduced activity measured for the KA variant in the context of the FIX-HSA fusion. Nonetheless, the introduction of the Padua amino acid substitution gives rise to the expected increase in specific activity, which largely compensates for the negative effect of the KA substitution.

Accordingly, the manuscript has been revised as follows;

Results (page 5, lines 100-106): “*To investigate the coagulant activity of the designed FIX-HSA fusions, aPTT-based and chromogenic activity assays were performed. The results showed dose-dependent coagulant activities, with normal (Fig. 1b) or hyperactive (Fig. 1c) profiles, dependent on the absence or presence of the Padua amino acid substitution (Supplementary Fig. 3). The introduction of the K5 substitutions resulted in either unaltered (K5R; 0.94-fold of FIX-HSA_{WT}) or reduced (K5A; 0.5-fold of FIX-HSA_{WT}) activity, while their combination with Padua gave rise to hyperactive features (averaging 9.49-fold higher than FIX-HSA_{WT}; Fig. 1b-d; Supplementary Table 1).*”

Discussion (page 12, lines 280-282): “*When K5A or K5R, which either weakens or strengthens the ability of FIX to bind Col4,^{30, 31} respectively, were introduced in the FIX-HSA fusion proteins, the FIX clotting activity was reduced or preserved, respectively.*”

Discussion (page 12, lines 282-286): “*Despite this, introduction of the Padua amino acid substitution largely compensated for the reduced activity associated with K5A, as hyperactive properties were observed for all Padua-containing fusion proteins.*”

New Supplementary Figure 3 with legend:

Supplementary Figure 3. Functional properties of FIX variants in unfused and HSA-fused formats. a) Secreted protein and activity levels of unfused FIX (left y-axis), either designed with the K5A (blue) or K5R (light purple) substitution, and the corresponding specific activity (right y-axis). Data represent the mean \pm SD ($n = 5$). Unpaired two-tailed Student’s t-tests were performed to study statistical difference (ns, not significant; *** $p < 0.001$; **** $p < 0.0001$). b) Specific activity of FIX_{WT} and Padua either unfused or fused to HSA_{WT} or HSA_{QMP}, and as WT, K5A, or K5R in position 5 of FIX. Results show the specific activity levels measured by an aPTT-based assay (no pattern) and a chromogenic activity assay (striped). Data represent the mean \pm SD ($n = 4$).

2. In Tg32 mice, HSA_{QMP} exhibited significantly higher MRT compared to HSA_{WT}, attributable to the hFcRn transgene. However, this difference was not observed in either FIXplus or HB mice. Moreover, in FIXplus and HB mice, the increase in MRT from Padua-HSA_{QMP} to PaduaKR-HSA_{QMP} was marginal, while the difference in MRT decreased further when comparing FIX-HSA_{WT} and FIXKR-HSA_{WT} in Tg32 mice (Table 2). Is there evidence suggesting that this phenomenon is linked to the binding affinity between mouse collagen IV and human FIX? Would

the use of human collagen IV potentially amplify the difference between Padua-HSA_{QMP} and PaduaKR-HSA_{QMP}?

We thank the reviewer for raising an interesting question related to the interplay between the albumin:FcRn and FIX:Col4 interactions as PK modulators.

As pointed out, the mouse models used provided results as a function of the FcRn/FIX setting, with experiments in Tg32 mice resulting in the highest MRT values for HSA_{QMP} variants. This is expected, and in line with the cross-species albumin:FcRn binding differences, as the plasma half-life of HSA is longer in mice expressing the human version of FcRn, than in those expressing the mouse receptor.

In FIX^{plus} and HB mice that allow comparison of different FIX settings, and thus Col4 accessibility, the increase in MRT for Padua_{KR}-HSA_{QMP} compared to Padua-HSA_{QMP} (FIX^{plus}, from 1.4 to 1.9, 36%; HB, from 1.2 to 1.4, 17%) was modest, as highlighted by the reviewer. Nevertheless, a trend in MRT increase between the two proteins was observed. This may relate to the presence (in FIX^{plus}) or absence (in HB) of competition between the infused molecule and the endogenous FIX. A similar competitive effect for FcRn binding has been previously observed by us in albumin KO mice, in which the difference in plasma half-life between HSA_{WT}- and HSA_{QMP}-fused FIX was less evident than in the presence of albumin as competitor (Lombardi et al. 2021, ref. 25 in the manuscript). Moreover, this is in line with clinical observations supporting that the CRM status affects the bioavailability of an infused FIX.

To further clarify, we have revised the Discussion on page 14 (lines 340-343), which now reads “Furthermore, the incremental improvements of the protein engineering steps were evident when correlating the MRT values achieved in the CRM⁺ (FIX^{plus} Balb/c) and CRM⁻ (HB) mouse models, which again highlighted the superior PK **properties and competitive advantage** of Padua_{KR}-HSA_{QMP}.” (changes highlighted in yellow).

We also agree that an additional layer of complexity might be due to a difference in binding of human FIX to the mouse or human forms of Col4, although to the best of our knowledge there is no reported data in this regard. It is worth noting that it has been postulated that local post-translational modifications of Col4 *in vivo* may affect binding to FIX (PMID: 27766061), however, this remains to be demonstrated. As such, we cannot exclude that interspecies differences may occur regarding post-translational modifications of human and mouse Col4 that may affect binding. However, if such species-specific differences exist, they will apply to all mouse models used in our study. Nevertheless, the KR/KA changes resulted in improved/decreased MRT in HB mice, as well as tissue presence in line with the engineering steps performed (Fig. 4i-1), further demonstrating the effects of improved or decreased Col4 binding in an *in vivo* context. To provide further insights, we have now performed both immunofluorescence (IF) and immunohistochemistry (IHC) studies on tissues harvested from the HB mice, with staining for both human FIX and mouse Col4. IF staining for human FIX revealed significantly higher signals in tissues of mice administered with Padua_{KR}-HSA_{QMP} than with Padua_{KR}-HSA_{QMP}, while the signals for Col4 were comparable. Moreover, IHC staining of tissues from the same organs were in line with the IF staining, with more staining achieved in tissues of mice administered with Padua_{KR}-HSA_{QMP} than with Padua_{KR}-HSA_{QMP}. Again, staining for Col4 showed comparable staining intensities irrespective of treatment. Thus, the new experiments further support that human FIX in the fusion proteins is capable of engaging mouse Col4 *in vivo* in mice, with an

effect of engineering in position 5. The new IF and IHC results are shown in new Figure 4m and Supplementary Figures 15-17.

Overall, the exploited mouse models with different FcRn/FIX settings, each of them potentially presenting intrinsic limits in term of PK evaluation of albumin-fused FIX, allowed us to provide experimental evidence even in the presence of a complex interplay between the albumin fusions and Col4, which impacts their plasma half-lives. We are conscious that only studies of the fusion designs in a “fully-human” HB context with mice engineered to be FIX deficient and transgenic for both the human forms of FcRn and Col4 would provide a full picture.

3. Why does Padua_{KA}-HSA_{QMP} exhibit lower MRT compared to FIX-HSA_{WT} in HB mice but not in FIXplus mice? Could the presence of CRM+ impact the clearance of Padua_{KA}-HSA_{QMP}?

The CRM+ status may modulate the PK properties of an infused FIX, as we have observed in HB patients with normal levels of circulating dysfunctional FIX (Branchini et al. 2021, ref. 43 in the manuscript). Based on the mouse studies, the plasma half-life of FIX is modulated by fusion to albumin, with HSA_{QMP} providing extended plasma half-life beyond that of HSA_{WT}, as well as by Col4-binding.

In this view, in HB mice, the negative impact of the K5A substitution affects MRT even in the HSA_{QMP}-containing scaffold, resulting in a value lower than that of the HSA_{WT}-bearing fusion protein. Conversely, in FIX^{plus} mice, the contribution of the HSA_{QMP}-driven improvement in plasma half-life might mask the impact of the K5A substitution, with a potential (additional or synergistic) contribution of the CRM+ status.

4. The specific activity of FIX is defined as the units of factor IX per milligram. It is recommended that FVIII activity (IU) levels are used instead of clotting time in figure 1b and 1c to avoid confusion.

We thank the reviewer for pointing this out. To avoid confusion, we have performed a new analysis of coagulant activity by comparing coagulation times with those of known concentrations (IU/ml) of Idelvion used as reference. The results showed dose-dependent coagulant activities, with normal or hyperactive profiles, dependent on the absence or presence of the Padua amino acid substitution.

Accordingly, these data have been added to new Figures 1b and 1c, and we have modified the text in Results (page 5, lines 100-103) that now reads “**To investigate the coagulant activity of the designed FIX-HSA fusions, aPTT-based and chromogenic activity assays were performed. The results showed dose-dependent coagulant activities, with normal (Fig. 1b) or hyperactive (Fig. 1c) profiles, dependent on the absence or presence of the Padua amino acid substitution (Supplementary Fig. 3).**”, as well as the corresponding figure legend.

Revised figure 1 with legend:

Figure 1. Design and activity profiles of FIX-HSA variants with modifications in both fusion partners. a) Four fusion proteins were designed with FIX genetically fused to HSA via a cleavable linker. FIX was modified in position 5 of the Gla domain (K5A or K5R) and in position 338 in the catalytic domain (R338L; Padua). HSA was either unmodified (WT) or engineered with three amino acid substitutions (E505Q/T527M/K573P; QMP) in DIII. Gla, γ -carboxyglutamic acid domain; EGF1/EGF2, epidermal growth factor like domains 1 and 2; AP, activation peptide. **b-c)** Coagulant activity of FIX measured by aPTT-based assays in FIX-deficient plasma supplemented with pure monomeric fractions of FIX-HSA_{WT} (**b**) or Padua-HSA_{QMP} (**c**). **d)** Specific clotting activity of each FIX-HSA variant. The data represent the mean \pm SD (n = 4), where specific activity is calculated as the ratio between activity and protein levels (FIX-HSA_{WT} = 1). Unpaired two-tailed Student's t-tests were used to study statistical difference (ns, not significant; * $p < 0.05$; ** $p < 0.005$; *** $p < 0.0005$; **** $p < 0.0001$). FIX-HSA_{WT}, black; Padua-HSA_{QMP}, gray; FIX_{KA}-HSA_{WT}, blue; FIX_{KR}-HSA_{WT}, orange; Padua_{KA}-HSA_{QMP}, pink; Padua_{KR}-HSA_{QMP}, purple. The illustration in panel a was created in Biorender.com.

5. Will any of these mutations or their combination have any impact on the immunogenicity of FIX protein?

We thank the reviewer for raising this question, which is relevant for clinical translation knowing that the already approved replacement therapies for hemophilia may, in some cases, elicit an immune response, and that the current study describes new designs engineered beyond the native FIX and HSA proteins.

To answer this question, we have performed two *in silico* prediction analyses to identify potential T- and B-cell epitopes, using the prediction tools NetMHC4.1 and BepiPred2.0. These are well-established prediction tools that we have used in previous published papers (Foss et al., Nature Com, 2024; Mester et al., mAbs, 2021). For comparison, we included the sequence for the approved FIX-albumin fusion albutrepenacog alfa (Idelvion; CSL Behring).

The results revealed that that it is unlikely that the amino acid substitutions introduced during the engineering steps would be associated with an increased immunogenicity, as a comparable number of potential epitopes were predicted for both the natural proteins and engineered designs including the commercial product albutrepenacog alfa. However, as we also have discussed in Mester et al., mAbs (2021), caution should be taken upon using such prediction tools due to largely unresolved challenges with such analyses (Greiff et al., Current Opinion in Systems Biology, 2020).

Based on these analyses, the following changes have been made in the manuscript to describe the *in silico* prediction results;

Discussion (page 12, lines 287-291), paragraph added: “*Regarding potential immunogenicity of the engineering strategies used to design the FIX-HSA fusions, we could not detect any warning increase in T- and B-cell epitopes using available prediction tools, as a comparable number of potential epitopes was shown for the FIX variants and the FIX-HSA fusions, including the approved clinical product Idelvion (Supplementary Tables 12 and 13). However, caution should be taken when using such tools, as discussed.*^{66, 67}”.

New Supplementary Table 12 (T-cell epitope prediction)

New Supplementary Table 13 (B-cell epitope prediction)

Methods (page 24, lines 635-639).

Minor issues:

- 1. Figure 4f cannot see the % remaining of Padua_{KR}-HSA_{QMP} in the figure. Is that because of overlapping with another group? Maybe using different symbols will help make the figure clearer.**
- 2. Figure 5f cannot see Padua-HSA_{QMP} group clearly. Is that due to overlap with FIX-HSA_{WT}? Again, maybe using different symbols will help make the figure clearer.**

We apologize if not all groups in Fig. 4f and 5f were equally visible. Indeed, in Fig. 4f, the Padua_{KR}-HSA_{QMP} overlaps with the curve for Padua-HSA_{QMP} making these two groups difficult to set apart, while in Fig. 5f, Padua-HSA_{QMP} and FIX-HSA_{WT} are overlapping.

In an attempt to adjust this, we have made a new Figure 4f where the lines have been made slightly thicker. We hope that this has made the data more visible to the reader.

Furthermore, in the new Figure 5f, the FIX activity in HB mice (compared to day 1) treated with Padua-HSA_{QMP} and FIX-HSA_{WT} are indistinguishable over time, hence the curves overlap. This is in line with the major aspect of this study, that these proteins engage the Col4 reservoirs comparably and do not distribute differently to tissues. Thus, to communicate this finding clearly, we have kept Fig. 5f as in the original manuscript.

REVIEWER #2

We would like to thank this reviewer for their contribution to the review process.

REVIEWER #3

The authors investigated the effect of different mutations of FIX on its pharmacokinetic and pharmacodynamic effects.

More in details it has been investigated the impact of engineering FIX for improved (K5R) or reduced (K5A) Col4 binding, on the pharmacokinetic 48 (PK) profile of hyperactive FIX Padua fused to an engineered human serum albumin (HSA; 49 HSA_{QMP}) exhibiting favorable neonatal Fc receptor (FcRn) engagement.

The results clearly demonstrated that Padua_{KA}-HSA_{QMP} showed a shorter half lifetime and activity compared to Padua_{KR}-HSA_{QMP} which has a much longer effect. This study paves the way for future therapeutic opportunities for optimization and personalization of HB replacement therapy.

Major comments

1. The authors should implement the data regarding the tissue accumulation of that Padua_{KA}-HSA_{QMP} compared to Padua_{KR}-HSA_{QMP}. They should perform IHC analysis from lungs, kidneys, liver, and knee joints tissue in order to show the localization of FIX compared to collagen IV. This analysis is essential to confirm the general hypothesis on the different binding of Padua_{KR}-HSA_{QMP} and Padua_{KA}-HSA_{QMP} to collagen.

We thank the reviewer for the comment and agree that IHC studies will allow us to study the tissue localization of FIX in light of Col4. In accordance, we have performed IHC studies on tissue sections from liver, lungs, kidneys, and knee joints harvested from HB mice (n = 3/group) which were administrated with Padua_{KR}-HSA_{QMP} or Padua_{KA}-HSA_{QMP}. The tissues were stained with anti-human FIX or anti-mouse Col4 antibodies. In agreement with data from the organ homogenates, tissue from mice treated with Padua_{KR}-HSA_{QMP} resulted in stronger staining signals for human FIX compared to those from mice treated with Padua_{KA}-HSA_{QMP}. When staining for mouse Col4, comparable staining intensities were observed irrespective of the fusion that was given. Thus, the new IHC data support that the K5R substitution results in an increased accumulation of the Padua-HSA_{QMP} fusion to Col4-containing tissues, compared to K5A. These data have been added to Supplementary Figure 16.

New Supplementary Figure 16 with legend:

Supplementary Figure 16. IHC staining against FIX and Col4 on tissues from HB mice. Immunohistochemical (3,3'-diaminobenzidine, DAB) staining of tissues from HB mice harvested at day 4 after treatment with Padua_{KA}-HSA_{QMP} or Padua_{KR}-HSA_{QMP}. **a)** Livers, kidney, lungs, and knee joints stained with an anti-human FIX antibody. **b)** Livers, kidney, lungs, and knee joints stained with anti-mouse Col4 antibody. Scale bar, 50 μ m; magnification, 40X.

Moreover, to provide further insights, we also performed immunofluorescence (IF) staining of the tissues and quantified the fluorescent signals at ten locations within each tissue. The results showed significantly higher fluorescent signals in tissues from HB mice treated with Padua_{KR}-HSA_{QMP} than from HB mice treated with Padua_{KA}-HSA_{QMP} (new Figure 4m). This was the case for all four tissues, consistent with the higher levels of Padua_{KR}-HSA_{QMP} detected in the organ homogenates (Figure 4i-l). Importantly, staining for mouse Col4 resulted in similar fluorescent signals between the groups and tissues (new Supplementary Figure 15).

Revised Figure 4 with legend:

Figure 4. K5R engineering of FIX-HSA increases plasma half-life and extravascular presence in HB mice. **a)** Illustration of the experimental setup in the study in HB mice, where 2.5 mg/kg of FIX-HSA was administered by I.V. injections and blood samples were taken daily until 4 days after. At termination (day 4), lungs, kidneys, liver, and knee joints were harvested and processed to homogenates, or paraffin embedded and prepared on slides before IF or IHC staining. Blood processed to plasma and tissue homogenates were analyzed by ELISA for protein quantification, and coagulant activity was measured in plasma by an aPTT-based assay. **b-d)** Protein concentration in plasma 1 hour, 24 hours, and 96 hours post-administration. Data are presented as the mean \pm SD (n = 4-5). Elimination curves showing **e)** protein concentration remaining in plasma over time and **f)** percentage of protein remaining in plasma at each time point compared to day 1 (= 100 %). **g)** Plasma half-life values shown as the mean \pm SD (n = 4-5). **h)** MRT values in FIX^{plus} Balb/c mice (x-axis) and HB mice (y-axis) analyzed in gPKPDSim. **i-l)** Homogenates of livers, kidneys, lungs, and knee joints sampled at termination were analyzed by ELISA for protein quantification. Data represent the mean \pm SD (n = 4-5) protein concentration. **m)** IF stained sections of liver, kidney, lung, and knee joint collected from HB mice 4 days after administration of Padua_{KA}-HSA_{QMP} or Padua_{KR}-HSA_{QMP} (green, anti-FIX; blue, DAPI for nuclei). Images show one representative animal per test article (scale bar: 100 μ m). The fluorescence signal at ten different fields per image was quantified. The data represent the mean \pm SD (n = 10). Unpaired two-tailed Student's t-tests were used to study statistical difference (ns, not significant; * $p < 0.05$; ** $p < 0.005$; *** $p < 0.0005$; **** $p < 0.0001$). FIX-HSA_{WT}, black; Padua-HSA_{QMP}, gray; Padua_{KA}-HSA_{QMP}, pink; Padua_{KR}-HSA_{QMP}, purple. The illustration in panel a was created in Biorender.com.

New Supplementary Figure 15 with legend:

Supplementary Figure 15. IF staining against Col4 on tissue from HB mice. IF staining (green, anti-Col4; blue, DAPI for nuclei) of tissues collected at day 4 from HB mice injected with Padua_{KR}-HSA_{QMP} or Padua_{KA}-HSA_{QMP} and quantification of the fluorescent signal from ten different fields per image. Padua_{KA}-HSA_{QMP}, pink; Padua_{KR}-HSA_{QMP}, purple. Scale bar, 100 μ m. ns, not significant.

The new IF and IHC results are now described in the revised manuscript as follows:

Results (page 9-10, lines 225-234): “To further study the tissue distribution of the fusion proteins, harvested organs were sectioned and stained with primary goat and rabbit polyclonal antibodies specific for human FIX or mouse Col4 before detection using species-matched secondary antibodies. Immunofluorescent (IF) staining revealed significantly stronger fluorescence intensity in tissues from HB mice treated with Padua_{KR}-HSA_{QMP} than with Padua_{KA}-HSA_{QMP} (Fig. 4m), while the signals for mouse Col4 were not significantly different (Supplementary Fig. 15). In line with this,

immunohistochemistry (IHC) staining confirmed detection of more Padua_{KR}-HSA_{QMP} than Padua_{KA}-HSA_{QMP} (Supplementary Fig. 16). Again, the tissue showed similar IHC staining of mouse Col4 irrespectively of treatment with the FIX-HSA variants (Supplementary Fig. 16). Notably, treating the tissues with only the secondary antibodies did not give any signal (Supplementary Fig. 17)."

Results (page 10, line 236): "*Thus, whereas K5A greatly reduces the extravascular presence of Padua-HSA_{QMP}, K5R increases its distribution to the liver, kidneys, lungs, and knee joints, at sites with mouse Col4 expression.*"

Discussion (page 13, lines 326-327): "*In agreement, significantly more Padua_{KR}-HSA_{QMP} than Padua_{KA}-HSA_{QMP} were detected in the same tissue when stained by IF and IHC.*"

Methods (page 22-24, lines 591-628)

2) Please provide the protocol number for authorization of animal studies.

As requested by the reviewer, we have added the protocol numbers in the revised text. The approval numbers are now listed in the method descriptions in the revised manuscript (page 21, lines 537-538, lines 551-552, and lines 562-563).

REVIEWER #4

The authors have reported on the in vivo distribution and functional characteristic of increased or decreased collagen binding potentials of albumin-fused gain-of-function hyperactive FIX (FIX-Padua) compared to WT. As the authors have also described, we also know that the efficacy of HB replacement therapy is evaluated based on the FIX:C levels in blood circulation plasma, while FIX which binds to extravascular type IV collagen appears to contribute to possible hemostasis and to affect the biodistribution of infused rFIX. In the present study, they investigated the impact of engineering FIX for increased (K5R substitution) or decreased (K5A substitution) type IV collagen binding potential mutant, on the PK profile of FIX-Padua fused to an engineered human albumin (HSA_{QMP}).

Their obtained results can be summarized as below:

1. Regardless of the K5 mutation, Padua-HSA, Padua_{K5R}-HSA, and Padua_{K5A}-HSA showed hyperactive properties and improved hFcRn binding and cellular recycling, resulting in extended half-life in hFcRn-transduced mice.

2. In mice mimicking CRM-positive patients expressing dysfunctional FIX, K5A-product negatively modulated the plasma half-life. In CRM-negative HB mice, K5A and K5R product exerted opposite effects on the PK and biodistribution profiles of the fusion.

Therefore, Padua_{K5A}-HSA showed no extravascular distribution, and the highest plasma levels at early time points. While, Padua_{K5R}-HSA_{QMP} showed increased extravascular distribution to the center of liver organ, and knee joints, and 3-fold longer functional half-life. The authors concluded that the impact of collagen binding potential on the PK profile of FIX fused to HSA, supporting the use of Padua_{K5A}-HSA and Padua_{K5R}-HSA as hyperactive short- or long-term therapeutics with opportunities for personalization of HB replacement therapy.

The concept and methodology of the present study appears to be reasonable and the obtained results may be enough and clear to explain the authors final conclusion. The development of tailored-made or personalized FIX product may be hoped. However, some major concerns can be raised in the present study.

1. Did the authors perform these biochemical or enzymatic analyses on the FIX-modified gain- or loss-of-function proteins generated in the present study? Did albumin-fused proteins completely albumin portion release and function as well as FIXa, when converted to FIXa? How about the properties of FIX/FIXa? Furthermore, did you investigate the properties of the intrinsic tease complex in vitro (association with FX or FVIIIa), binding potential to phospholipid membrane, and reaction with FXIa or FVIIa etc.? These results would be necessary because this is stated as a future therapeutic application that the authors conclude.

We thank the reviewer for their focus on the biochemical assessment of the engineered fusions. To study the functional features of FIX after activation of both unfused and fused formats, we evaluated the specific coagulant activity of the FIX-HSA_{WT} and Padua-HSA_{QMP} fusion variants compared to the respective unfused FIX molecules. The results showed that fusion to either HSA_{WT} or HSA_{QMP} did not affect the specific activities of FIX_{WT} or Padua, respectively, except for when engineered with the K5A substitution, which resulted in decreased specific activities (New Supplementary Figure 3)- Nonetheless, Padua engineering of FIX resulted in an enhanced specific activity of the fusion protein (on average 9.49-fold enhanced activity compared to FIX-HSA_{WT}).

Moreover, to provide insights into the impact of the KA amino acid substitution on coagulant activity, we evaluated the ability of Padua-HSA_{QMP}, Padua_{KA}-HSA_{QMP}, and Padua_{KR}-HSA_{QMP} to bind phospholipids, as suggested by the reviewer. To do so, an ELISA-based assay was designed, where plates were coated with phospholipids (MP-Reagent, DAKO) and serial titrations of the proteins were added, before bound fusion proteins were detected using an anti-HSA antibody (Bethyl Laboratories). The results showed a reduced binding capacity of Padua_{KA}-HSA_{QMP} compared to Padua_{KR}-HSA_{QMP} and Padua-HSA_{QMP} (New Supplementary Figure 5). These findings may contribute to explain the reduced activity of the KA-containing variants.

Finally, knowing that release of albumin from a FIX-HSA fusion is important for efficient coagulant activity, we performed a time-course incubation with plasma-derived activated factor XI (pdFXIa), a physiologic FIX activator that would also cleave at the rationally designed linker site (Lombardi et al. 2021, ref. 25 in the manuscript) leading to albumin detachment. With all the data presented above in our hands, we chose to study Padua_{KR}-HSA_{QMP} and Padua_{KA}-HSA_{QMP} in these experiments. Incubation of pure fractions of the fusion proteins with FXIa resulted in a comparable efficiency in the rate of albumin release over time, as shown below in new Supplementary Figure Figure 4. Thus, the KA and KR engineered FIX-HSA fusion proteins are cleaved by pdFXIa comparably, resulting in efficient release of albumin.

New Supplementary Figure 3 with legend:

Supplementary Figure 3. Functional properties of FIX variants in unfused and HSA-fused formats. **a)** Secreted protein and activity levels of unfused FIX (left y-axis), either designed with the K5A (blue) or K5R (light purple) substitution, and the corresponding specific activity (right y-axis). Data represent the mean \pm SD (n = 5). Unpaired two-tailed Student's t-tests were performed to study statistical difference (ns, not significant; *** p < 0.001; **** p < 0.0001). **b)** Specific activity of FIX_{WT} and Padua either unfused or fused to HSA_{WT} or HSA_{QMP}, and as WT, K5A, or K5R in position 5 of FIX. Results show the specific activity levels measured by an aPTT-based assay (no pattern) and a chromogenic activity assay (striped). Data represent the mean \pm SD (n = 4).

New Supplementary Figure 4 with legend:

Supplementary Figure 4. Time-course evaluation of albumin detachment from FIX-HSA fusions at the cleavable linker site upon cleavage by FXIa. **a-b)** Western blotting analysis of Padua_{KR}-HSA_{QMP} (**a**) or Padua_{KA}-HSA_{QMP} (**b**) incubated with pdFXIa in a time-course. Zymogen fusions (125 kDa) or detached albumin (67 kDa) were detected by polyclonal anti-HSA antibodies. A schematic illustration of the respective protein fragments is shown on the right. **c-d)** Densitometric analysis of band intensity, indicated as % at each time point, of Padua_{KR}-HSA_{QMP} (**c**) or Padua_{KA}-HSA_{QMP} (**d**) as zymogen fusion protein (grey circles) or detached HSA (red circles). A schematic illustration of the fusion protein structure with the cleavable site within the linker sequence indicated is shown to the right.

New Supplementary Figure 5 with legend:

Supplementary Figure 5. *In vitro* phospholipid binding of Padua-HSA_{QMP} fusion proteins engineered in position 5 of FIX. Results from an *in vitro* ELISA-based phospholipid binding assay where phospholipid vesicles were coated in wells and concentration gradients of Padua-HSA_{QMP} (gray), Padua_{KA}-HSA_{QMP} (pink), and Padua_{KR}-HSA_{QMP} (purple) were added, before the HSA-region of the fusion proteins was detected by an anti-HSA antibody. The data represent the mean \pm SD (n = 2).

In the revised manuscript, we have included the following revisions related to this reviewer question;

Results section (page 5, lines 106-109): “*Moreover, upon incubation of FIX with plasma-derived activated FXI (pdFXIa), HSA was released from Padua_{KA}-HSA_{QMP} and Padua_{KR}-HSA_{QMP} via its cleavable linker at comparable rates (Supplementary Fig. 4).*”

Discussion (page 12, lines 284-286): “*Importantly, HSA was efficiently cleaved from both the K5A- and K5R-containing fusions upon activation by FXIa, which is required for optimal coagulant activity of FIX.*”²⁵”

Methods (page 16-17, lines 420-428)

2. In the present study, FVIII:C was measured by the one-step method, but was it evaluated by a chromogenic assay?

In our study, we provide results on coagulant activity in plasma measured through the one-stage method, an assay being closer to a physiological context, even from a clinical stand-point, and widely used for *in vitro* and *in vivo* functional studies. Nevertheless, in the revised manuscript, we have included an evaluation of the activity of the HSA_{WT}- and HSA_{QMP}-containing variants performed using a chromogenic assay (Biophen, Aniara Diagnostica; previously used by us, ref. 79 in the manuscript). This assay showed, for each protein scaffold, that the K5R substitution did not have an impact on the activity of FIX, however, decreased activity was observed for the K5A-bearing fusion variants and the corresponding reference, in either WT (FIX_{WT}) or Padua. We also included the unfused FIX (FIX_{WT}) and Padua variants as controls, to provide evidence of preserved properties, either normal or hyperactive, upon albumin fusion (New Supplementary Figure 3).

The results from the chromogenic assay were consistent with the results from the aPTT-based assay (New Supplementary Figure 3).

New Supplementary Figure 3 with legend:

Supplementary Figure 3. Functional properties of FIX variants in unfused and HSA-fused formats. a) Secreted protein and activity levels of unfused FIX (left y-axis), either designed with the K5A (blue) or K5R (light purple) substitution, and the corresponding specific activity (right y-axis). Data represent the mean \pm SD ($n = 5$). Unpaired two-tailed Student's t-tests were performed to study statistical difference (ns, not significant; *** $p < 0.001$; **** $p < 0.0001$). **b)** Specific activity of FIX_{WT} and Padua either unfused or fused to HSA_{WT} or HSA_{QMP}, and as WT, K5A, or K5R in position 5 of FIX. Results show the specific activity levels measured by an aPTT-based assay (no pattern) and a chromogenic activity assay (striped). Data represent the mean \pm SD ($n = 4$).

3. Related to the above comment, since this is a mouse-based experiment, all coagulation factors are mouse-based coagulation proteins, and since FIX is a human FIX mutant protein, the differences between mouse-derived coagulation factor reactions and human-derived coagulation protein reactions should be fully considered.

We do agree that when assessing activity in mouse plasma a potential confounding effect might be present due to the different species-specific proteins. However, the presence of mouse plasma is an unavoidable condition shared in all *in vivo* settings. Importantly, we would like to highlight that we measured similar specific activity between purified variants both when evaluated through aPTT-based assays in human FIX-deficient plasma (Figure 1d) and in plasma deriving from HB mice containing mouse, and not human, coagulation factors (Fig. 5e).

4. Although the current study was evaluated with FIX activity measurement, real data on global coagulation potential and hemostatic potential are also needed to lead to your conclusions from this study. Data on comprehensive coagulation measurement using for example, ROTEM/TEG and thrombin generation, and hemostatic potential measurement such as tail-cut bleeding should also be evaluated.

We do agree that evaluation of the hemostatic potential is relevant in a translational perspective of the present study. As such, we have in the revised manuscript included results from a tail clip bleeding assay performed in HB mice ($n = 5$) (new Figure 5i). With the 3R principle in mind, we selected Padua_{KR}-HSA_{QMP} for this experiment.

By comparing blood loss upon tail clipping with the activity levels of the Padua_{KR}-HSA_{QMP} variant in mouse plasma, we measured a concentration-dependent correction of the hemorrhagic phenotype, as indicated by the inverse correlation, shown in new Figure 5i. We believe that these data, obtained by studying the long-term therapeutic candidate (Padua_{KR}-HSA_{QMP}), provide insights into the hemostatic impact of the Padua_{KR}-HSA_{QMP} variant and strengthen the relevance of the functional data and the associated therapeutic potential.

New Figure 5 with legend:

Figure 5. The K5R amino acid substitution grants FIX-HSA with enhanced activity and extended functional half-life in HB mice. a-c) FIX activity in plasma of HB mice after 1 hour, 24 hours, and 96 hours post-administration of a FIX-HSA fusion. Data represent the mean ± SD (n = 4-5) compared to the activity in plasma of mice that received FIX-HSA_{WT}. **d)** Change in FIX clotting activity in plasma over time. Data represent the mean ± SD (n = 4-5) and are presented as the percentage of that of FIX-HSA_{WT} (= 100%). **e)** Specific activity measured in mice over time, calculated as the ratio between FIX clotting activity and protein concentrations relative to that of FIX-HSA_{WT} (= 1). Data represent the mean ± SD of each animal at all time points (n = 5). **f)** Change in FIX activity in plasma over time. Data represent the mean ± SD (n = 4-5) and are presented as the level of clotting activity over time compared to one day post-administration (day 1 = 100 %). **g)** Functional half-life of the FIX-HSA variants measured by gPKPDsim. Data represent the mean ± SD (n = 4-5). **h)** Correlation between MRT values obtained by applying the FIX plasma concentration (x-axis) and FIX clotting activity (y-axis) to gPKPDsim. Data represent the mean ± SD (n = 4-5) and were applied to a simple linear regression. **i)** Inverse correlation between Padua_{K_R}-HSA_{QMP} activity and hemoglobin loss upon tail clipping at day 4 post-administration. Data represent the mean ± SD (n = 5) and were applied to a simple linear regression. Unpaired two-tailed Student's t-tests were used to study statistical difference (ns, not significant; * p < 0.05; ** p < 0.005; *** p < 0.0005; **** p < 0.0001). FIX-HSA_{WT}, black; Padua-HSA_{QMP}, gray; Padua_{K_A}-HSA_{QMP}, pink; Padua_{K_R}-HSA_{QMP}, purple.

In the revised manuscript, we have included the following revisions related to this reviewer question;

Results (page 11, lines 265-268): “To evaluate the ability of the fusion protein to promote hemostasis in vivo, a tail clip assay was performed on HB mice 4 days after treatment with Padua_{K_R}-HSA_{QMP}. The results showed an inverse correlation between blood loss and plasma activity levels (Fig. 5i), confirming that the Padua_{K_R}-HSA_{QMP} present in blood efficiently promoted hemostasis.”

Discussion (page 14, lines 343-346): “Importantly, the decrease in blood loss as a function of Padua_{K_R}-HSA_{QMP} activity indicates a concentration-dependent correction of the disease phenotype, thus underscoring the importance of functional activity and its relevance in therapeutic potential.”

Methods (page 22, lines 574-582).

5. The present study mentioned the focus point of CRM+ in terms of extravascular depot of endogenous FIX. However, the mechanism induced CRM+ pattern is not only about this mechanism. It is not appropriate to discuss CRM+ as a whole only in terms of extravascular depot of endogenous FIX. It seems too much to say that it is indicated for treatment.

We thank the reviewer for pointing this out, and have accordingly modified the discussion around this as such;

Discussion (page 12, lines 300-302): “**In fact, the CRM+ status may modulate the compartment bioavailability of an infused FIX by potential competition with dFIX.**”

Discussion (page 13, line 312): removed “**and thus be beneficial especially for the HB patients who are CRM+**” (strikethrough and highlighted in yellow in resubmitted manuscript).

Discussion (page 14, line 339): removed “**independent of CRM status**” (strikethrough and highlighted in yellow in resubmitted manuscript).

6. What is the immunogenicity of the FIX modified proteins which were generated this time? If it is supposed to be applied to hemostatic therapy, I think this analysis should be fully analyzed. for example, in silico analysis etc.

We thank the reviewer for bringing up this important aspect. To provide further insights to the potential immunogenicity of the new fusion protein designs, we have performed two *in silico* analyses of potential T- and B- cell peptide epitopes, using the prediction tools NetMHC4.1 and BepiPred2.0. The sequences for all fusions as well as the individual fusion partner and the approved FIX-HSA fusion protein albutrepenacog alfa (Idelvion; CSL Behring), were included in the analyses.

The results revealed no increase in immunogenicity upon introduction of the K5A or K5R substitutions in FIX, neither when compared to the non-engineered albumin fusion protein nor the approved fusion. However, caution must be taken when using such prediction tools, as previously discussed in Mester et al. *Mabs* (2021) and in Greiff et al., *Current Opinion in Systems Biology* (2020).

The *in silico* analyses are included in the resubmitted manuscript:

Supplementary Table 12 (T-cell epitope prediction)

Supplementary Table 13 (B-cell epitope prediction)

Discussion (page 12, lines 287-291), paragraph added: “**Regarding potential immunogenicity of the engineering strategies used to design the FIX-HSA fusions, we could not detect any warning increase in T- and B-cell epitopes using available prediction tools, as a comparable number of potential epitopes was shown for the FIX variants and the FIX-HSA fusions, including the approved clinical product Idelvion (Supplementary Tables 12 and 13). However, caution should be taken when using such tools, as discussed.^{66, 67}**”

A method description is given in Methods (page 24, lines 635-639).

Kristin et al. demonstrated different strategies for changing activity or half-life of FIX protein using a combination of hyperactive mutant FIX (Padua), enhanced or impaired affinity to Col4: K5R and K5A mutation, and engineered HSA which displayed enhanced affinity to FcRn. These strategies can dramatically impact the pharmacokinetics of FIX protein. This study also showed the effect of binding ability to Col4 with K5 mutations *in vivo*. The increasing of extravascular distribution of K5R mutation can enhance the functional half-life. The K5A mutation also showed the highest FIX level at early time points and can be used for on-demand treatment with an immediate need of FIX levels in plasma. Additionally, the authors also provide the half-life data of EHL FIX protein in Tg32 transgenic mice, FIX^{plus} mice and HB mice to mimic the CRM+ or CRM- HB patients. In summary, the combination of the hyperactive mutation (R338L) in FIX_{Padua}, enhanced or decreased binding to Col4 (K5R and K5A, respectively), and improved engagement with HAS (HSA_{QMP}) can significantly enhance the efficacy of replacement therapy for HB for different purposes. This new EHL FIX protein (Padua_{KA}-HSA_{QMP} and Padua_{KR}-HSA_{QMP}) can be used in either short- or long-term therapeutics according to the requirement.

The data overall are well presented, and the paper is generally well written. Notably, FIX_{Padua} is recognized as a hyperactive FIX mutation, and HSA_{QMP} enhances FcRn binding affinity, substantially improving the half-life of FIX, as demonstrated in Tg32 mice, which possess hFcRn to mimic the human condition. One question is why FIX K5 engineering has a lesser effect on FIX MRT when comparing K5 (WT) and K5R (K5 vs. K5R) in HB mice (1.2 vs. 1.4) than in FIX^{plus} mice (1.4 vs. 1.9). Considering that HB mice are CRM- phenotype and their extracellular depots are not saturated by non-functional FIX protein, the discrepancy in MRT between K5 and K5R should be more pronounced than in FIX^{plus} mice. If the varying binding affinity between human and mouse collagen IV to human FIX is speculated as one of the reasons or the authors have different perspective, those points should be addressed in the discussion section.

Major issues:

1. Why did FIX with the K5A mutation (Padua_{KA}-HSA_{QMP} or FIX_{KA}-HSA_{WT}) exhibit lower clotting activity (Table S1) and specific activity compared to K5 (Padua-HSA_{QMP} or FIX-HSA_{WT}) and K5R (Padua_{KR}-HSA_{QMP} or FIX_{KR}-HSA_{WT})? (Figure 1d). Did the KA mutation affect FIX activity?
2. In Tg32 mice, HSA_{QMP} exhibited significantly higher MRT compared to HSA_{WT}, attributable to the hFcRn transgene. However, this difference was not observed in either FIX^{plus} or HB mice. Moreover, in FIX^{plus} and HB mice, the increase in MRT from Padua-HSA_{QMP} to Padua_{KR}-HSA_{QMP} was marginal, while the MRT decreased further when comparing FIX-HSA_{WT} and FIX_{KR}-HSA_{WT} in Tg32 mice (table 2). Is there evidence suggesting that this phenomenon is linked to the binding affinity between mouse collagen IV and human FIX? Would the use of human collagen IV potentially amplify the difference between Padua-HSA_{QMP} and Padua_{KR}-HSA_{QMP}?

- Why does PaduaKA-HSA_{QMP} exhibit lower MRT compared to FIX-HSA_{WT} in HB mice but not in FIX^{plus} mice? Could the presence of CRM+ impact the clearance of PaduaKA-HSA_{QMP}?
- The specific activity of FIX is defined as the units of factor IX per milligram. It is recommended that FVIII activity (IU) levels are used instead of clotting time in figure 1b and 1c to avoid confusion.

- Will any of these mutations or their combination have any impact on the immunogenicity of FIX protein?

Minor issues:

- Figure 4f cannot see the % remaining of PaduaKR-HSA_{QMP} in the figure. Is that because of overlapping with another group? Maybe using different symbols will help make the figure clearer.

2. Figure 5f cannot see Padua-HSA_{QMP} group clear. Is that due to overlap with FIX-HSA_{WT}? Again, maybe using different symbols will help make the figure clearer.